# La Palma landslide tsunami: computation of the tsunami source with a calibrated multi-fluid Navier-Stokes model, impact assessment, and model intercomparison.

Stéphane Abadie[1], Alexandre Paris[1,2], Riadh Ata[3], Sylvestre Le Roy[4], Gael Arnaud[5], Adrien Poupardin[2,6], Lucie Clous[1], Philippe Heinrich[2], Jeffrey Harris[3], Rodrigo Pedreros[4], and Yann Krien[5]

[1]Universite de Pau et des Pays de l'Adour, E2S UPPA, SIAME, Anglet, France
[2]CEA, DAM, DIF, Arpajon 91297, France
[3]LHSV, Ecole des Ponts, CEREMA, EDF R et D, Chatou, France
[4]BRGM, Orléans, France
[5]Université des Antilles, Laboratoire LARGE, Campus de Fouillole, 97157 Pointe-a-Pitre, Guadeloupe
[6]Institut de Recherche en Constructibilité, Université Paris-Est, ESTP Paris, 28 avenue du Président Wilson, 94230, Cachan, France

**Correspondence:** Stephane Abadie (stephane.abadie@univ-pau.fr)

**Abstract.** In this paper, we present new results on the potential La Palma collapse event, previously described and studied in Abadie et al. (2012). Three scenarios (*i.e.*, slide volumes of 20, 40 and 80 km³) are considered, modeling the initiation of the slide to the water generation using THETIS, a 3D Navier-Stokes model. The slide is considered as a Newtonian fluid whose viscosity is adjusted to approximate a granular behavior. After 5 minutes of propagation with THETIS, the generated
5  water wave is transferred into FUNWAVE-TVD, after 15 minutes of Boussinesq model simulation, to build a wave source suitable for propagation models. The results obtained for all the volumes are made available through a public repository. In the present paper, this signal is then propagated with the Boussinesq model FUNWAVE-TVD, taking into account dispersive effects, to provide reference simulation results and allow studying impact on France and Guadeloupe. Although the slide modeling approach applied in this study seemingly leads to smaller waves compared to former works, the wave impact is still very
10 significant for the maximum slide volume considered on surrounding islands and coasts, as well as on remote most exposed coasts such as Guadeloupe. In Europe, the wave impact is significant (for specific areas in Spain and Portugal) to moderate (Atlantic French coast). The tsunami propagation is also performed using three other models for the purposes of comparison. While this exercise demonstrates the role of physical dispersion in this particular case, a proper model resolution appear to be the critical aspect to ensure accurate results.

**Keywords:** Tsunamis, Atlantic Ocean, Numerical modeling, Volcanic hazards and risks

## 1 Introduction

Recent catastrophes due to exceptionally strong tsunamis (Athukorala and Resosudarmo, 2005; Mikami et al., 2012) have called the need for extensive tsunami hazard assessment or reassessment in several countries (e.g., National Tsunami Hazard

Mitigation Program (NTHMP) in the USA (Tehranirad et al., 2015), or the Tsunamis in the Atlantic and the English ChaNnel Definition of the Effects through numerical Modeling (TANDEM) project for France (Hebert, 2014)). In this context, the hazard associated to various potentially tsunamigenic sources has to be evaluated. This work usually covers the most frequent sources, namely co-seismic displacements and submarine landslides, but long-return period sources, like volcano tsunami sources, must also be investigated. Volcanic islands may indeed have the potential to generate tsunamis (see for instance the recent case of Anak Krakatau (Paris et al., 2019; Grilli et al., 2019)), even mega-tsunamis, through a flank collapse process (Tappin et al., 2019), known to occur relatively regularly (Elsworth and Day, 1999). Footprints of such gigantic past events are large underwater landslide debris surrounding specific oceanic islands (Masson et al., 2002) and marine conglomerates at high elevation on the flanks of other ones (Paris et al., 2018). Unfortunately, the tsunami hazard associated to volcanic islands is very difficult to determine due to the complexity of the processes involved as well as uncertainty of the associated return period. Nevertheless, although likely very rare, these events may have such dramatic consequences that they should be taken into account in extensive hazard assessment studies. The present paper is an attempt, in the framework of the previously cited TANDEM project, to assess the potential impact on France, some parts of Western Europe, and remote French territories (*i.e.*, the archipelago of Guadeloupe) of a tsunami generated by an hypothetical collapse of the Cumbre Vieja volcano at La Palma Island (Canary Islands, Spain).

This volcano has drawn a strong interest among the scientific community since the first alarming work published on that case (Ward and Day, 2001). There have been several attempts to numerically simulate the waves generated by the Cumbre Vieja collapse. The first work (Ward and Day, 2001) was severely criticized (Mader, 2001; Pararas-Carayannis, 2002) due to the allegedly extreme landslide volume considered and the linear wave model used. In more recent computations, Gisler et al. (2006) used a 3D compressible Navier-Stokes model to simulate the slide and the resulting wave. An extrapolation of near field decay led the authors to conclude, as in Mader (2001), that this wave height would not represent such a serious threat for the East coast of North America or South America. Starting from the Gisler et al. (2006) near field solution, Løvholt et al. (2008) simulated the transoceanic propagation of the tsunami source with a Boussinesq model, therefore including dispersive effects. The propagation is shown to be very complex due to the combined effects of dispersion, refraction, and interference. The authors also found smaller waves than Ward and Day (2001), but still potentially dangerous for the U.S. coasts. Abadie et al. (2012) proposed a similar approach but based on a 3D multi-phase incompressible Navier-Stokes model to simulate the landslide and the generated wave. Because of the likelihood uncertainty, they proposed four different sliding volumes, ranging from 20 to 450 km$^3$, obtained from a former slope stability study. The impact of these potential sources on U.S. coasts was studied in Tehranirad et al. (2015) in the framework of the NTHMP, with propagation computed using the FUNWAVE-TVD model. In the far-field, the generated tsunamis appear to be made of wave trains of 3 to 5 long-crested waves of 9 to 12 min period. If the wave height appears very significant along the 200 m isobath (in the range of 20 m) for the largest volume considered, a strong decay is also observed due to bottom friction on the continental shelf. Moreover, besides the initial directionality of the sources, coastal impact is mostly controlled by focusing/defocusing effects resulting from the shelf bathymetric features. Based on the same source and methodology, but an inundation computed using a refined shallow water

model, Grilli et al. (2016) found the CVV to cause the largest impact among possible far-field sources, with up to 20 m runup at the critical sites for the 450 km$^3$ scenario.

Computations performed by Gisler et al. (2006) or Abadie et al. (2012) were both based on inviscid or quasi-inviscid slide flow. In the present paper, the computations carried out in Abadie et al. (2012) are redone, improving their accuracy by calibrating the slide fluid viscosity in order to approach a granular slide (Sections 2.1 and 3.1) with a Newtonian model. Then, the same filtering process as in Abadie et al. (2012) is applied with the new wave sources to produce a wave signal which can be propagated by depth-averaged models (Sections 2.4 and 3.2). The three wave sources are then propagated using FUNWAVE-TVD (Section 2.2.1) and the results in the Caribbean Sea, in Western Europe and in France (Section 3.3) analyzed.

One of the goal of the TANDEM program was also the comparison of the models developed or used by the different partners of the project namely: Calypso developed by the French Alternative Energies and Atomic Energy Commission (CEA), Telemac2D developed by Electricité De France (EDF) R&D group , Funwave-TVD used by the Bureau de Recherches Géologiques et Minières (BRGM) group and SCHISM by Université des Antilles. Here we take the opportunity of this case study to compare models on a real case and analyze the differences. The interest is double. This project involves partners who are already in charge of tsunami hazard assessment while others may play a role in this field at the national level in the future. The first interest is to provide an inter-comparison of the codes used at in the different institutes. This comparison will be valuable for future operational use. On the other hand, this comparison is made on a real case, therefore including all the inherent complexity and uncertainties (bathymetry, mesh, numerical parameters, physical parameters, etc.) usually associated to a practical case. Such a comparison is rarely attempted in usual benchmark exercises which focus more frequently on specific processes such as run-up, tsunami generation, etc., in order to make the interpretation easier. Nevertheless, even though the analysis is not straightforward because models are not based on the same assumptions, numerical methods, mesh types, a comparison including all the complexity may also be of interest as it allows to judge all the effects at once and potentially lead to practical recommendations valuable for future studies. Therefore the originality of this comparison on a real case is the second interest of this part of the study. Accordingly, the rest of the study is organized around a comparison of the different model results (see Sections 2.2.2 for description and 3.4 for the results comparison). Finally, tsunami impact is assessed in different areas in Section 3.5, and results interpreted and discussed in Section 4.

## 2 Method

### 2.1 Navier-Stokes simulation of wave source

The model used for wave source computations is the Navier-Stokes multi-fluid model THETIS already described in Abadie et al. (2010) and Abadie et al. (2012) in the context of waves generated by landslides. In this 3D model, water, slide and air are simulated based on the incompressible Navier-Stokes equations for Newtonian fluids. The interfaces between phases are tracked using the Volume-Of-Fluid (VOF) method. The same set-up as in Abadie et al. (2012) is used in this study, so the reader is referred to this former work to find more details on the model.

The $\mu(I)$-rheology (Jop et al., 2006) has also been implemented in THETIS to model dry dense granular flows and been validated by comparing with a dry granular column collapse (Lagrée et al., 2011). The three material-dependent parameters are $I_0$, $\mu_s$, and $\Delta\mu$. They define the friction coefficient, $\mu(I)$, which only depends on the inertial number, $I$. In THETIS, these variables are evaluated on each point of the slide, and the viscosity $\eta$ is computed and imposed as the local fluid viscosity value in the NS equations. This gives a viscosity in the slide that is space and time-dependent. In the present work, we used the usual values found in the literature for the model parameters, namely: $\mu_s = 0.43$, $\Delta\mu = 0.39$, and $I_0 = 0.27$. Note that this formulation is, so far, only valid for a dry collapse (Clous and Abadie, 2019), and is therefore only used here as a reference for the initial motion.

THETIS belongs to the immiscible multi-phase full Navier-Stokes type of solver. It has been validated against several benchmark cases involving tsunami generated by 2D and 3D solid blocks (Abadie et al., 2010), and granular subaerial and submarine slides (Clous and Abadie, 2019). As such, it is more sophisticated with respect to the slide motion than models such as the SAGE model (Gisler et al., 2006), which rely on a compressible formulation of the equations or the 3D Navier-Stokes model described in (Horrillo et al., 2013), which employed a simplified VOF method, taking advantage of the large aspect ratio of the tsunami waves. Other recent models of interest regarding landslide tsunami generation include the NHWave model described in (Ma et al., 2015; Kirby et al., 2016; Grilli et al., 2019) which is a two-layer Sigma coordinates model for granular landslide motion and surface wave generation with a depth-averaged description of the slide and a 3D non-hydrostatic tsunami wave. For submarine landslides involving cohesive visco-plastic soils, the model BingClaw (Løvholt et al., 2017; Kim et al., 2019) based on a non-linear Herschel–Bulkley model, incorporates buoyancy, hydrodynamic resistance and remolding, which appear crucial to properly represent the underwater landslide dynamics. The latter model has been used to study the dynamics of the Storegga Slide about 8000 years ago as well as the 1929 Grand Banks landslide and tsunami. Finally, Eulerian-Eulerian two-phase models such as the one described in Si et al. (2018b) and Si et al. (2018a) are very promising approaches able to describe the flow within the grains as well as the grain/grain interactions but their applicability to practical cases has not been demonstrated yet.

As previously mentioned, the tsunami sources proposed in Abadie et al. (2012) were computed based on Navier-Stokes simulations using a Newtonian fluid of very low viscosity (quasi-inviscid) for the slide. In 2D preliminary tests, the generated waves were shown to increase gradually when lowering the slide viscosity. So the simulations performed in Abadie et al. (2012) represent the worst case possible with this model for a given slide volume. In the present paper, the aim is to propose a more realistic source prediction by calibrating the previous Navier-Stokes model with respect to recent experimental measurements of waves generated by granular slides. The experimental results considered are: Viroulet et al. (2013) (see also Viroulet et al. (2014)) for subaerial slides, and Grilli et al. (2017) for submarine slides.

Viroulet et al. (2013) conducted a 2D physical experiment with glass beads in order to represent an equivalent granular slide. This experiment was carried out in a flume of dimensions 2.20 m long, 0.4 m high, and 0.2 m wide. The beads were placed initially above water on a 45° slope as in the Figure 2. Glass beads had a density of 2500 kg·m$^{-3}$ and a diameter of 1.5 mm in the first case, 10 mm in the second. Water depth was 14.8 cm and 15 cm for the first and second case, respectively. Four gauges monitored the surface elevation at $x_1 = 0.45$ m, $x_2 = 0.75$ m, $x_3 = 1.05$ m and $x_4 = 1.35$ m.

In the numerical model, the slide is modeled as a fluid with a Newtonian rheology. A simulation with a $\mu(I)$-rheology was also performed for comparison purpose on the same configuration as Viroulet et al. (2013). Nevertheless, except the latter simulation, the rest of the simulations presented in this study with THETIS was carried out with a Newtonian rheology and a calibrated viscosity.

The space and time steps are $\Delta x = 5$ mm, $\Delta y = 2$ mm and $\Delta t = 10^{-3}$ s, respectively. The flow is solved with the projection algorithm and a VOF-Total Variation Diminishing (TVD) interface tracking is performed.

For the first experimental case presented in Viroulet et al. (2013), simulations with different values of viscosity were carried out. Figure 3 compares the height of the first wave at the four gauges. The wave simulated with the lowest viscosity, as in Abadie et al. (2012), appears to be almost twice as high as the experimental results. This first result shows the need to consider a better calibration of the model to produce more realistic results in the La Palma case. The first wave and the wave train which follows are well reproduced for a viscosity of 10 Pa·s, even if the slide at this viscosity is shown to be slower than in the experiment. The same overall behavior is observed in the second case, with glass beads diameter of 10 mm, but a higher value of viscosity has to be set in order to fit the experimental wave heights. Note that the slide motion simulated is still slower than in the experiment. This may be due to the one-fluid model formulation, which does not allow for the flow to pass through the granular medium as in reality. Energy transfers from slide to free surface, not detailed in the present study, were computed based on numerical results (Clous and Abadie, 2019) and show that waves are generated extremely quickly in this subaerial experiment. This is certainly why the differences observed in slide velocity after some time do not induce large wave discrepancies.

The first benchmark case was also simulated with the $\mu(I)$ rheology. The results show that the wave height is quite close to the experimental results. Comparing to the computation with the Newtonian fluid, during the first 0.5 s, where the waves are generated, the equivalent viscosity calculated with $\mu(I)$-rheology is homogeneous within the slide volume and close to the best Newtonian case. Therefore, this simulation shows that a well-calibrated Newtonian rheology can be used to model a complex granular rheology at least in this specific case for which energy transfers are very fast. This will be the approach used in the present paper.

The experiment presented in Grilli et al. (2017) was also simulated using THETIS. The experiment consisted of 2 kg of 4 mm glass beads released underwater over a slope of 35° in a water depth of 0.330 m. The slide was modeled as a Newtonian fluid, first with parameters defined in Grilli et al. (2017), $i.e.$, a viscosity of 0.01 Pa·s and a density of 1951 kg·m$^{-3}$. A few other viscosity values were also tested to evaluate the sensitivity of the model. The results show that with a slide viscosity of 0.01 Pa·s, the first wave is higher than the experimental value and the wave train is not correctly reproduced on the first gauge. By reducing the viscosity, the generated waves are lower. We observe that with a viscosity of 1 Pa·s, the first wave is close to the experimental results as well as the first waves in the wave train. Overall the results on wave height appears satisfactory while the slide is still slower than in the experiment.

To extrapolate these results for the La Palma computations, the following reasoning is adopted. First, it is assumed that the real slide is well represented by the granular medium used in the experiment. This approach is not deterministic as there are

important differences between this experiment and the real case but at least it may be considered as a better assumption than the worst case scenario presented in Abadie et al. (2012).

Second, the 2D cross section of the La Palma slide in Abadie et al. (2012) is $\sim$8 km$^2$ compared to $\sim$4 for Viroulet's slide extrapolated at real scale. As these surfaces are of the same order, the slide dynamics are assumed to be roughly similar. Third, the La Palma slide is partially submerged but with a larger subaerial portion. Because of this, the real case would be more similar to the first experiment (Viroulet et al., 2014) than to the second one (Grilli et al., 2017).

The equivalent viscosity for the real case is then obtained by scaling the optimal viscosity obtained after calibrating the model against the experiments. Froude and Reynolds numbers should be the same at reduced and real scales leading to:

$$\frac{u}{\sqrt{gh}} = \frac{u'}{\sqrt{gh'}} \tag{1}$$

$$\frac{\rho u h}{\mu} = \frac{\rho u' h'}{\mu'} \tag{2}$$

where $g$ is the acceleration of gravity, $u(u')$ a characteristic velocity, $h(h')$ a characteristic length scale and $\mu(\mu')$ the equivalent viscosity at real scale (reduced scale respectively). Combining the two equations leads to:

$$\frac{\mu}{\mu'} = \sqrt{\frac{h^3}{h'^3}} \tag{3}$$

which for a viscosity $\mu' = 10$ Pa·s at reduced scale gives $\mu = 4.4 \times 10^7$ Pa·s at real scale given the length ratio. The slide considered in Abadie et al. (2012) (Figure 1) being partially submerged, the latter viscosity value is arbitrarily reduced to $\mu = 2 \times 10^7$ Pa·s to take into account of the result obtained with Grilli et al. (2017)'s experiment.

Based on these hypothesis, simulations were performed with three initial slide volumes corresponding to 20, 40 and 80 km$^3$, respectively. The largest slide volume considered in Abadie et al. (2012), namely 450 km$^3$ is not considered in this paper (see section 4).

## 2.2 Models used for long distance propagation

### 2.2.1 Main model: FUNWAVE-TVD

As dispersive effects are expected to play a significant role in this case (Løvholt et al., 2008), the reference model for this study is the Boussinesq model FUNWAVE-TVD.

FUNWAVE-TVD, run here for long-distance propagation by the BRGM group, is the most recent implementation of the Boussinesq model FUNWAVE (Wei et al., 1995), initially developed and extensively validated for nearshore wave processes, but equally used to perform tsunami case studies. The FUNWAVE-TVD code, which solves the Boussinesq equations of Chen (2006) with the adaptive vertical reference level of Kennedy et al. (2001), with either fully-nonlinear equations in a Cartesian framework (Shi et al., 2012) or a weakly-nonlinear spherical coordinate formulation with Coriolis effects (Kirby et al., 2013). It uses a TVD shock-capturing algorithm with a hybrid finite-volume and finite-difference scheme to accurately simulate wave

breaking and inundation by turning off dispersive terms (hence solving the Non-Linear Shallow Water (NSW) equations during breaking) once wave breaking is detected (detection based on the local wave height). The code is fully parallelized using the Message Passing Interface (MPI) protocol and efficient algorithms allowing a substantial acceleration of the computations with the number of cores. For operational uses, FUNWAVE-TVD has received many convenient features, such as the use of nested

grids to refine the simulations in the interest areas, or the use of heterogeneous Manning coefficients to characterize bottom friction. For the transatlantic simulations here, the Manning coefficient is a constant $0.025$ m$^{-1/3}$·s.

In the framework of the U.S. NTHMP program, FUNWAVE-TVD has been validated for both tsunami propagation and coastal impact, through an important set of analytical, laboratory and field benchmarks (Tehranirad et al., 2011). Other recent applications have allowed the validation of the model on real cases, such as the Tohoku-Oki tsunami (Grilli et al., 2013).

The simulation of the propagation of the tsunami to the coastlines was performed with nested grids (Figures 4 and 5) from 2.7-km resolution (Atlantic Ocean) to 930 m (Antilles), 450 m (north Atlantic area), 310 m (Guadeloupe Archipelago), 110 m (Aquitaine region) and 20 m (Gironde estuary).

### 2.2.2   Other models used

As formerly pointed out in the introduction, we also used three other different models for the simulation of the propagation of

the tsunami, each of them run by different institutions. These models use the NSW equations, except for one version of one of the model (*i.e.*, Calypso, see below). We compare the models with each other and also with the reference model, FUNWAVE-TVD. More specifically, the objective of this comparison is to try to assess the part of uncertainty due to the use of different numerical methods solving the same equations (i.e., here NSW equations) and the one due to inclusion or not of dispersive effects. The presentation of the other models used follows.

**Calypso**: Calypso is a code developed by CEA and used for tsunami propagation (Poupardin et al., 2017; Gailler et al., 2015). The user can choose to solve either the non-dispersive (NSW) or dispersive (Boussinesq model following Pedersen and Løvholt (2008)) non-linear long wave equations, written in spherical coordinates. A Crank-Nicolson scheme for the temporal discretization and a finite-difference scheme for spatial derivatives are used to solve both NSW and Boussinesq equations. For the Crank-Nicolson scheme, an iterative procedure enables the solving of the implicit set of equations. The convergence

criteria is applied to the continuity equation. The spatial discretization uses centered differences for linear terms as well as for advection terms. For Boussinesq equations, the implicit momentum equations are solved by alternating implicit sweeps in the $x$ and $y$ components using an Alternating Direction Implicit (ADI) method. For a given direction, the dispersion terms in the other direction are discretized explicitly. For each direction ($x$ and $y$), a tridiagonal system of equations is then solved at each iteration, following Pedersen and Løvholt (2008). The numerical scheme of Calypso has been described in Poupardin et al.

(2018).

Four levels of nested grids are used in this computation (Figures 4 and 5). The mother grid covers Canary Islands and a large part of the Atlantic Ocean to the French coasts. It is a 2-km resolution grid with a total of $1351{\times}1298$ cells. The second grid of $1294{\times}1404$ cells covers all the French Atlantic Ocean coastline and the north of Spain with a 500-m resolution. Four grids are used to simulate the propagation of water waves in coastal regions: the so-called "Brittany" grid covers a large region

in the south of Brittany with a 125-m resolution; the "Gironde" grid covers the mouth of the Gironde estuary with a 125-m resolution; the "Saint-Jean-de-Luz" grids with a first grid of 125-m resolution and a smaller one of 32.5-m resolution which covers the bay of Saint-Jean-de-Luz in the southwest of France.

In the main simulation performed, denoted as Calypso-B-NSW, the offshore propagation was simulated by using the Boussi-nesq model to take into account the dispersive effects in the Atlantic Ocean, then NSW equations are solved in the daughter grids in order to reduce the computation time. The wave impact assessment is realized using this mixed method for the French coasts and calculating run-up with wet and dry conditions. Finally, to better assess the actual effect of dispersive terms and taking advantage of the possibility to turn those effects on or off in Calypso, a full NSW (denoted as Calypso-NSW) and a full Boussinesq (denoted as Calypso-B) simulations were performed to the west and to the east of the La Palma source.

**Telemac-2D**: Telemac-2D is the 2D component of the Telemac-Mascaret system (http://www.opentelemac.org). It is a finite element and a finite volume solver based on the resolution of NSW. For this paper, it was run by the EDF R&D group. The non-conservative form of the NSW equations is used for the discretization of the finite-element kernel. In Telemac-2D, the equations are solved in exactly the same as for Cartesian coordinates, however using a local estimation of the latitude, an element-by-element Mercator projection is used to provide a distance correction (Hervouet, 2007). Moreover, Coriolis terms are taken into account for these simulations. Therefore proper transoceanic propagation can be considered in current version of the model. Telemac-2D is massively parallelized using an MPI domain decomposition approach. This feature is very useful for the simulation of large problems such as the one presented here. Finally, the inundation process is not calculated in this simulation as the wetting-drying algorithm was turned off and no breaking effect modeled.

In this work, the mesh used in Telemac-2D has 12.5 million triangular elements and 6.4 million nodes. The model resolution ranges from 10 km off-shore to 700 m near the coasts with a cell expansion ration of 1.15. The limits go from the Senegal coasts in the South to the Arctic circle in the North and from the eastern American coasts to the European ones. Figure 4 shows the computational domain used with Telemac-2D. All the boundaries are set as solid walls (note that the first wave does not reach the boundaries at the end of the simulation time).

**SCHISM**: Semi-implicit Cross-scale Hydroscience Integrated System Model (SCHISM) (Zhang et al., 2016), is a derivative product of the Semi-implicit Eulerian-Lagragian Finite-Element (SELFE) model (Zhang and Baptista, 2008a). It is run here by Université des Antilles. Although the code is able to solve the 3D Reynolds Averaged Navier-Stokes equations in hydrostatic or non-hydrostatic mode, in this study only one sigma layer is used and equations are depth integrated leading to 2D NSW equations with additional source terms for Coriolis effect, bottom friction dissipation and horizontal eddy viscosity in the momentum equation.

A first set of simulations have been performed over the whole domain starting using the wave signal obtained 15 min after the volcano collapse as input. In these first simulations, the NSW equations are solved over an unstructured mesh that covers the part of the Atlantic basin between the Canary Islands and the Lesser Antilles arc (Figure 4). In order to avoid projection errors, a spherical coordinates option, based on Comblen et al. (2009) approach has been used in the present study. The resolution is adapted to be able to accurately reproduce wave trains of period of 12 min or more, with at least 20 nodes per wavelength in deep ocean. This gives a resolution of 4000 m in the deepest part of the domain and around 1800 m near the Canary archipelago.

A second set of simulations has been performed to compute the tsunami inundation along the Guadeloupe coastline (section 2.5).

## 2.3 Locations of numerical output

West in the vincinity of the Canary archipelago, a first synthetic gauge (Gauge 1) is used to analyze the wave at the beginning.

In the Caribbean Sea, we detail the tsunami waves features in the vicinity of the Guadeloupe Archipelago. The latter is located 61°W and 16°N in the Lesser Antilles at 4600 km from the Cumbre Vieja Volcano. It is made up of four main groups of islands (Figure 7) with a total surface of 1628 km$^2$. Four synthetic gauges are used in this area: Gauges 8 and 9, respectively north and south of Guadeloupe Island, Gauge 10 northeast of the Caribbean arc and Gauge 11 east of Guadeloupe Island.

In Europe, the following synthetic gauges are used (Figures 4 and 5): Gauge 2 south of Portugal and Spain to evaluate the impact in this region, Gauge 3 in the French abyssal plain, Gauge 4 in the continental shelf off the French Atlantic coast, and Gauges 5, 6 and 7 located on the French coastline (in front of the Gulf of Morbihan, near the Gironde estuary and at the entrance of the Saint-Jean-de-Luz bay). The locations, coordinates and depths of the eleven gauges are provided in Figures 4, 5 and 7 and in Table 1.

## 2.4 Transition from Navier-Stokes to propagation models

As noted in the original THETIS simulations presented in Abadie et al. (2012), the landslide, as modeled, continues to move for a very long time (more than half an hour), but the slide local Froude number is super-critical for only a short time (less than 100 s), and it is only during this super-critical period when the resulting tsunami wave continues to grow significantly. As a result, it is not necessary to model the entire slide run-out in order to capture the generation of waves that will affect distant shorelines.

Taking the result from the THETIS model after 300 s of simulated time, once several wave fronts have already propagated away from the generation site, integrating velocity over depth, we transfer the state of the model to the Boussinesq wave model FUNWAVE-TVD. However, the water around the still-moving slide includes highly turbulent three-dimensional effects that cannot be represented correctly in a Boussinesq model. To remove the residual flow (that is not expected to generate significant waves) near the slide, we apply an *ad hoc* filter, as determined by numerical experimentation. It consisted in multiplying the output of THETIS (*i.e.*, free surface elevation and each velocity component) by a spatially varying function, removing the interior flow while keeping a smooth initial condition for FUNWAVE. This function is Gaussian, with a standard deviation of 15 km and the center is located at coordinates ($-10$ km, $-10$ km). For more details, including validation of this approach, see Abadie et al. (2012).

After this filter is applied, local Boussinesq wave modeling is conducted on a 500-m resolution bathymetric grid taken from the Global Multi Resolution Topography (GMRT) (Ryan et al., 2009). In order to take advantage of the fully nonlinear version of FUNWAVE-TVD, a Cartesian coordinate grid system is used. To project this onto the local area, a transverse secant Mercator projection is used (similar to the UTM system, but centered at 28.5°N and 18.5°W corresponding to ($+68$ km, $+14$ km)). The distortion of the entire grid is less than 1%.

After this initial phase of propagation, the results of wave elevation and horizontal velocity are transferred to larger-scale simulations to predict impact on various coastlines, as detailed in Section 2.2.

## 2.5 Tsunami impact assessment

Independently of the wave signal quality, an accurate assessment of the impact of a given tsunami also requires refined computations on nested refined grids including local friction coefficients and an accurate knowledge of the bathymetry and the topography. In the present study, this extensive work was performed in Guadeloupe. For this archipelago, the transoceanic propagation is performed using the code FUNWAVE-TVD while nearshore propagation and inundation is carried out with SCHISM. The two grids FUNWAVE-TVD and SCHISM cover the same domain (Figure 4). A hotstart is made from the wave train of the FUNWAVE-TVD grid over the SCHISM grid at t=18900 s (*i.e.*, 5 h 30 min after the volcano collapse). At this time, the first wave is about 180 km Eastward from La Desirade. For these specific simulations, along the Guadeloupe coastline, and for the aerial part where specific features may obstruct the water flow inland, resolution reaches 10 m. Inundation process relies on a specific inundation algorithm that is detailed and benchmarked in Zhang and Baptista (2008b). The Manning coefficient is adjusted as a function of the land use as shown in Figure 6. For the submerged area, 10 classes of Manning values were used while 50 classes have been used for the aerial domain based on Corine Land Cover dataset (Büttner et al., 2004). In order to avoid reflection along the domain limit, boundary conditions are set to Flather type (Flather, 1976).

Maximum flow depth is also provided in this paper for three areas of the French Atlantic coastline : Morbihan, Gironde Estuary and Saint Jean de Luz bay. For these areas a refined computations was performed with Calypso-B-NSW, and the tsunami impact assessed through the maximum surface elevation at the isobath -5 m, taking advantage of the high-resolution topo-bathymetric data provided by the Service Hydrographique et Océanographique de la Marine (SHOM) institute in the framework of the TANDEM project.

## 3 Results

### 3.1 Wave source computation

Figures 8 and 9 provide the complete sequence of the computed slide contours, thicknesses and related water surface elevations for the 80 $km^3$ scenario, obtained in (Abadie et al., 2012) and in the present computation considering a viscous flow with a viscosity of $2 \times 10^7$ Pa.s. With a higher viscosity, the slide dynamics and the resulting wave generation changes significantly compared to the inviscid case for the 80 km$^3$ volume case. The slide is much slower, more compact and regular in shape during the energy transfer to water surface.

The bulge, which was very developed in the previous case (Figure 10), is scarcely noticeable, although it still exists (Figure 11). The slide tip is also slower ($\sim$30 m·s$^{-1}$, Figure 11(b)) compared to the original simulation ($\sim$100 m·s$^{-1}$, Figure 10). The rear part of the slide, where the velocity is maximum, is still very fast ($\sim$120 m·s$^{-1}$) at the initial stage of the process (Figure 11(a)) but then the maximum velocity decreases to about 50 m·s$^{-1}$ (Figures 11(b) and (c)).

As a consequence of lower velocity and slide cross section reduction, the wave train generated is significantly less energetic than in the inviscid case (Figures 8 and 9). Nevertheless, the sequence of wave formation shows similarities with the generation of a first free surface positive elevation reaching 400 m in the new case (compared to 800 m previously) at t=90 s, which then exhibits radial decrease and frequency dispersion. Additionally, the very large depression of the mean sea water level, observed at the end of the wave generation process in the previous inviscid case, is less visible in the new simulation.

For this volume, after almost 10 minutes of propagation, the leading wave, which was previously about 80 m high, only reaches $\sim$30 m in the new viscous slide case (Figure 12(c)). Note that the wave energy focus has the same direction in both cases (*i.e.*, 20° south of West).

Figure 9 is not repeated for smaller slide volumes (*i.e.*, 20 and 40 km$^3$) as the slide evolution and wave train formation sequence show very similar pattern as compared to the 80 km$^3$ case.

Nevertheless, there is a significant variation of wave amplitude depending on the slide volume considered (Figure 13). At t=5 min, the leading wave is $\sim$80 m high in the largest slide volume scenario (*i.e.*, 80 km$^3$) and is only 50 m and 20 m for smaller slide volumes (40 and 20 km$^3$, respectively).

## 3.2 Filtered solution

Taking the THETIS solution (Figure 13) after the initial 5 minutes of propagation, and applying the filter described in Section 2.4, the subsequent wave propagation is simulated with the Boussinesq wave model FUNWAVE-TVD with a 500 m grid for an additional 15 minutes, which is sufficient to consider the interaction between the tsunami and the nearby islands.

The effect of the filtering can be seen clearly in Figure 14, where flow near La Palma is strongly damped, but the leading waves are unaffected. As shown by Abadie et al. (2012), this has been found to better represent the first several wave fronts and the overall wave field, as compared to an unfiltered solution.

The potential dispersive character of the wave train can be assessed by investigating the frequencies present in the wave spectrum. To that purpose, the wave signal close to the source in the direction of the maximum wave energy and the associated Fourier transform is presented in Figure 16. At this location, the depth is 4432 m. Linear waves can be considered as shallow water waves if their respective wave length $L$ verifies $L > 20h$. Still considering linear wave theory, this condition is only met for wave periods less than 4 min in this particular depth. Hence, the wave energy included in the frequency band 1 min to 4 min, which is obviously not negligible in Figure 16, can be considered as a superposition of intermediate water waves whose celerity depends of the period, not only on depth. For this part of the spectrum, which represents approximately 25% of the overall wave train energy, dispersion is expected to occur during the next propagation phase. Note that the frequency band concerned with dispersion will evolve during the propagation with depth increase or decrease.

The resulting wave elevation and velocity fields (e.g., Figure 15) is used as initial condition in FUNWAVE-TVD with larger grids for predicting impact on the distant coastlines. Similar simulations are also carried out with Calypso, Telemac-2D and SCHISM.

## 3.3 Propagation: FUNWAVE-TVD Results

Figures 17 and 18 show the maximal simulated sea surface elevation for the 80 km$^3$ scenario computed by FUNWAVE-TVD at an oceanic scale, from the source to the studied areas. A gradual decrease of the maximum wave height due to radial attenuation can be observed, modulated by energy focusing in narrow directions as already pointed out in Løvholt et al. (2008).

Territories close to the generation area are highly affected. The first locations impacted are the other surrounded Canary Islands, nearby archipelagos (Madeira Island, Cape Verde) and west Africa, especially western Sahara (Dakhla city - 100,000 inhabitants) and specific parts of Morocco by refraction on shallower part of the local bathymetry (Agadir, Essaouira, Safi - 800,000 inhabitants overall). In the latter areas, the waves are larger than 5 m.

The wave propagating toward Europe is obviously less energetic than in the western direction on which the main part of the energy is focused (Figure 17). Nevertheless, Portugal, the western coast of Spain and to a lesser extent, the southern coasts of Ireland and England are significantly affected. Lisbon, Porto, Vigo and Corunna appear to be the main cities at risk for the 80 km$^3$ tsunami scenario with a surface elevation of about 2 m. When approaching the French Atlantic coastline (Figure 18), the wave experiences shoaling on the continental shelf and the wave height slightly increases. Even though France is less affected than the previous territories, as the coasts are protected by the Iberian Peninsula, waves reach up to 1 m at various points located north of the Gironde Estuary up to the northern part of the Brittany peninsula.

Figure 19 shows the free surface signal in several selected points (Figures 4, 5 and 7) for the 80 $km^3$ scenario. The surface elevation reaches 0.75 m at Gauge 2 between the south of Portugal and the north of Morocco (followed by a though of the same amplitude) and around 0.15 m at Gauge 3 in the abyssal plain of the Bay of Biscay. These results are approximately consistent with a $r^{-1}$ propagation attenuation. Gauge 4 is located right after the beginning of the continental shelf. The increase of wave height is not very significant compared to Gauge 3 due to the large wavelength. Closer to the coast, the wave shoaling appears more significant with waves reaching about 0.40 m in south Brittany, 0.25 m in the Gironde estuary, and 0.40 m in Saint-Jean-de-Luz. Taking the first free surface increase as indicator, respective tsunami arrival times are 1 h 30 min, 2 h 50 min, 3 h 30 min, 4 h 15 min, 4 h 4 min and 3 h 50 min of propagation, respectively at Gauges 2 to 7.

Nearby Guadeloupe (Figure 19 (G) and (H)), the waves reaching the coasts are still significant with a first elevation of 0.75 m at Gauge 8 and 0.5 m at Gauge 9. Note that the second wave, which also features a large through appears to be the largest in this area.

The frequency content of the wave signal for the 80 km$^3$ scenario at the different gauges is also shown on Figure 19. As expected, due to dispersion, waves involving periods smaller than 4 min, whose respective celerity is smaller, are no longer visible in the spectrum whatever the gauge considered. In front of the French Atlantic coast and in South Brittany (*i.e.*, Gauge 4 and Gauge 5, respectively), the signal is made up of two main frequency bands respectively centered on 10 min and 40 min. The propagation toward the southern parts of the Bay of Biscay also shows a gradual decrease of the energy fraction associated to the highest frequencies (*i.e.*, T<30 min). Hence, in Saint-Jean-de-Luz or in the Gironde Estuary, the signal is dominated by waves between 30 min to 40 min periods with also some energy remaining in the lower frequencies (mainly 100 min). This is probably due to the fact that only the largest wavelengths are able to refract enough to reach these locations. We also note

that the very low frequency wave signal component (T>200 min) present in Gauges 2 and 3, is decreased in Gauges 4, 5 and 6 and not present in Gauge 7. For Guadeloupe (Gauges 8 and 9), compared to the wave signal close to the source (*i.e.*, Gauge 1), high frequencies involving periods less than 10 min are no longer observable and the signal is mainly composed of waves between 10 min and 100 min period. This is probably a manifestation of dispersion as during the transoceanic propagation, the wave train meets several time depths larger than 6000 m.

The wave train generated by the 20 km$^3$ slide (Figure 20) shows very similar frequency spatio-temporal evolution with less energy and no low frequency motion (*i.e.*, T>100 min). [Note that we observe a lag time of 5 to 10 minutes of the arrival times between the two slide scenarios.] The case of 40 km$^3$ is not presented as its characteristics can be deduced from the two former ones.

## 3.4 Propagation: Comparison with other models

Figure 21 shows a comparison of the free surface signal computed at Gauge 6 by the reference model FUNWAVE-TVD and Calypso for different grids (*i.e.*, (A) coarse grid only (Figure 4), (B) nested computation: coarse + intermediate grid, (C) coarse + intermediate + fine grid (Figure 4 and Figure 5). We recall here that Calypso was run in Boussinesq mode on the coarsest grid and in shallow water mode (non-dispersive) in the finer grids. The solutions computed by the two models on the finest resolution appear very similar at least for the three first waves (Figure 21 (C)). The comparison of panel (A), (B) and (C) gives an idea of the model convergence in the context of nested computations. On that particular point, the solutions computed by Calypso show less differences with grid resolution than FUNWAVE-TVD. For instance, the wave signal obtained with the intermediate grid is already close to the one obtained with the finest grid. This is not the case of FUNWAVE-TVD, which, with the coarsest grid, shows a wave signal with a clear cut in the high frequencies also visible in the spectra (Figure 21 A').

The results obtained with the same code (Calypso), but turning off (*i.e.*, Calypso-NSW) or on (*i.e.*, Calypso-B) the dispersive terms are presented on Figure 22. In Gauges 3 and 4 located in Europe, the spectra show slight differences over the whole frequency band. The signals computed by the NSW version also contain more energy in the high frequencies. In the direction of maximum energy in the far field (*i.e.*, Gauges 10 and 11 near Guadeloupe), the differences between the two simulations only appear on first half of the spectrum involving high frequencies. We also observe a slight delay between the two signals, the Calypso-NSW signal being a little in advance compared to the one obtained with Calypso-B.

Figures 23 and 24 now propose comparisons between NSW models (Telemac-2D and SCHISM, respectively) and Boussinesq models (Calypso-B and FUNWAVE-TVD, respectively). In Figure 23, Telemac-2D and Calypso-B are compared in Gauges 3 and 4. Here, the differences between the models results appear mainly on the frequency content. The arrival times are the same, the signals are in phase, but the Calypso-B simulation obviously contains more high frequencies than the Telemac-2D one. The shape of the spectra in the low frequency band are more similar in shape although the Telemac-2D is more energetic in this frequency band.

In Figure 24, SCHISM and FUNWAVE-TVD are compared in the Guadeloupe area after a transoceanic propagation. Again arrival times computed by both models agree very well. Overall, the correspondence between both models results is surprisingly good as also evidenced by the spectra correspondence.

## 3.5 Tsunami impact assessment

Figure 25 shows the distribution of the maximum surface elevation in the most refined domains of Calypso-B-NSW. A running average on 10 points has been applied to present more readable results. The flow depth at -5 m is in average 1 m in the Morbihan area with a one specific location (latitude 47.3 °N) submitted to a large 3 m flow depth. In the Gironde estuary and Saint-Jean-de-Luz areas, the flow depths are less than 1 m except in north part of the Gironde estuary and the south part of Saint-Jean-de-Luz where about 1 m flow depth is found.

For the Guadeloupe Archipelago, Figure 26 shows the spatial distribution of the maximum surface elevation for the town of Sainte-Anne (a) and the town of Saint-François (b). The extent of inundation illustrates the potential dramatic consequences and the need for evacuation of town centers. Incoming waves may reach several meters at the shore line, threatening the fisheries facilities of Sainte-Anne and the district of La Coulée in Saint-François. Urban areas are particularly exposed such as Saint-François, les Saintes, Sainte-Anne or Le Moule. As a consequence, the 80 km$^3$ scenario should be considered as a major tsunami with catastrophic consequences.

Regarding the 20 km$^3$ scenario (inundation maps not shown here), the overall flooded surface would reach about 9 km$^2$ and therefore this event should be already considered as an important tsunami event with an appropriate warning and evacuation of beaches, seafront, and close shore areas. In the 40 km$^3$ scenario (inundation maps not shown here), the flooded surface may reach 22 km$^2$ including potentially dense urban areas such as Saint François or Terre-de-Haut in les Saintes.

## 4  Discussion

The main goal of the present study was to improve the state of the art for the potential La Palma tsunami source and to use this new proposed scenario to perform an impact assessment for Europe and particularly for French territories. Such high return period events with potentially catastrophic consequences are particularly important to study as accurately as possible since, due to the difficulty to assess their precise return period, they often serve as reference for hazard mitigation study (Tehranirad et al., 2015).

The first result of the present work is the new tsunami source computed by Navier-Stokes simulation (for the initial 5 minutes), *ad hoc* filtering and Boussinesq wave propagation (for the following 15 minutes). As stressed previously, this source is more realistic than that considered in Abadie et al. (2012) due to the much larger viscosity used which is assumed to better approximate a granular slide. To support this, a comparison with existing granular experiments was performed, and the results extrapolated at real scale using a Froude/Reynolds similitude. Based on this new computation, we observed a significant diminution of the initial wave compared to the first assessment proposed in Abadie et al. (2012) (*i.e.*, wave height approximately half that of previously computed after 10 minutes of propagation for the 80 km$^3$ scenario). The new source (after filtering and propagation in the Boussinesq model) as well as comprehensive data on the slide are made available through the SEA scieNtific Open data Edition (SEANOE) portal Abadie et al. (2019). This data allow potential users to either compute the slide on their own and do the whole sequence of computation, or start from the already filtered wave solution to carry out propagation and impact studies.

The second result is a presumably better impact assessment in Europe generally, and a new detailed impact assessment for France and Guadeloupe. Considering a credible yet extreme 80 km$^3$ scenario, it is shown that the impact on the French Atlantic coast would remain moderate, but could also be significant on the coast of Portugal and be very significant in the Guadeloupe Archipelago. A direct comparison with Tehranirad et al. (2015) is difficult as the areas of interest were not the same in the two papers. Nevertheless, for instance, Tehranirad et al. (2015) found waves up to 10 m in the vicinity of Western Sahara, and 5 m waves on the Portuguese coast while they respectively reach 5 m and about 2 m in the present work, so the decrease is clear also far from the source.

An additional product of the study is the comparison between several numerical models in regions of overlapping interest, either based on dispersive or non-dispersive set of equations. While more complete benchmarks are considered within international or national project such as NTHMP (Horrillo et al., 2015) or the present TANDEM project, this comparison on a large-scale problem is able to provide some interesting physical insight. Tsunamis are generally considered as non-dispersive waves which can be satisfactorily approximated by the NSW equations. Nevertheless, this assumption is often not valid for tsunami generated by landslide (e.g., Mader (2001)), due to the much shorter wavelength. In the case of the La Palma collapse scenario, Gisler et al. (2006) and Løvholt et al. (2008) emphasized the importance of the dispersive effects, even in the far-field. In the present study, we recall that FUNWAVE-TVD is dispersive, Calypso-B-NSW included dispersive effects for the offshore propagation but not in the nearshore areas, whereas SCHISM and Telemac-2D were run in a non-dispersive manner.

Several results are obtained from this comparison with a few contradictions. First, the analysis of the wave signal obtained with FUNWAVE-TVD close to the source confirmed the presence of high frequency waves prone to dispersion in the depths encountered in this area of the Atlantic Ocean. Hence, physically, dispersion is expected and theoretically an appropriate Boussinesq modeling is required. The results obtained with FUNWAVE-TVD appear consistent with what is physically expected, high frequency waves progressively disappearing from the spectra during the propagation. The comparison between FUNWAVE-TVD and Calypso-B-NSW, which showed a good agreement, allowed to mutually validate the models and secure the results obtained (even though some discrepancies remain in the low frequency band). The methodology of performing transoceanic simulation in Boussinesq mode and shifting to NSW mode in the nearshore area is also validated through the good match observed in Figure 21 (C). This figure also stresses the effect of resolution in tsunami propagation simulations. Indeed, such computations are generally CPU expensive and the mesh is often adapted to this constraint, but Figure 21 shows that the results largely vary with resolution. Therefore, convergence of the results is also a critical aspect to verify and demonstrate in order to obtain accurate results. In the present study, both Boussinesq models are found to converge approximately toward the same solution which appears encouraging. We also took advantage of the possibility to run Calypso model in two modes: with or without the dispersive terms. The results obtained with this simulation, shown in Figure 22, allows to clearly quantify the role of dispersive effects for this particular case. Changes mainly in the high frequency band are observed in the spectra.

The comparisons carried out between the Boussinesq models and the pure NSW models are more difficult to interpret as they somehow contradict the previous results. To that respect, Telemac-2D results are quite surprising as, based on Figure 22, we would expect more energy in the high frequency band in the Telemac-2D NSW model than in the Boussinesq Calypso model whereas the contrary is obtained. To our opinion, this kind of surprising results may illustrate the role of resolution in space

and time and the need to obtain results convergence in order to allow proper models comparison. This is a mandatory first step, maybe more important than considering dispersive terms. The comparison between SCHISM and FUNWAVE-TVD is also quite unexpected. Here, very similar results are obtained after transoceanic propagation with a dispersive and a non-dispersive model. The explanation may be found by considering the large distance of propagation considered in this case and the fact that all the NSW models are somehow also affected by numerical dispersion inherent to the spatial discretization used. It is possible that this source of dispersion allows to get rid of the highest frequencies along the large distance travelled to end up with a very comparable signal as the one obtained with FUNWAVE-TVD.

Of course there are some limitations in this study which may provide the basis for future improvements.

First, this study should not be considered as a hazard assessment *stricto-sensu* because the return period aspect is not considered and the sensitivity in the landslide parameters not covered extensively. For a review on Probabilistic Tsunami Hazard Analysis (PTHA) methods, the reader is referred to Grezio et al. (2017) for instance. Instead, the current study presents plausible particular scenarios based on state-of-the-art numeral models. Note that the Navier-Stokes model, which provides interesting information for this kind of processes, is still too heavy to be employed in PTHA computations.

Second, we used a glass beads based experiment (Viroulet et al., 2013) to calibrate the Navier-Stokes simulation of the La Palma slide. If this is an improvement compared to the very coarse inviscid initial estimation (Abadie et al., 2012), which should be more considered as a worst case, such a laboratory experiment still is a huge simplification of the complexity expected in a real volcano collapse. An accurate description of such a complex process at real scale is still beyond the capabilities of current models. Therefore, there is here a very important source of uncertainty which the reader has to be aware of and this uncertainty propagates and affects the impact results. Furthermore, this work is not an hazard study which could have been performed, for instance, by considering different values of slide viscosity but at much higher computational cost. The position of this paper is rather to give an illustration of what could be expected from such an event by presenting results at least consistent with the current state of the art in terms of laboratory experiments and therefore propose an improvement compared to the previous published results on that case.

The present work did not explicitly take into account the possibility of a retrogressive scenario. Whether the flank collapse occurs en masse or in successive stages is obviously crucial in terms of wave generation. In this study, we proposed several slide volume scenarios which can be used for a crude assessment of the wave reduction in case the collapse occurred as several separate events with no interactions between the successive slides (e.g., the 20 km$^3$ scenario may give an idea of what would happen if a 80 km$^3$ slide were occurring progressively or in sequence). The interactions could be left for future research even though field evidences tend to show that these collapses may have occurred as separate events (Wynn and Masson, 2003) rather than in an actual retrogressive way.

On the other hand, the extreme scenario of 450 km$^3$ as studied in (Ward and Day, 2001; Løvholt et al., 2008; Abadie et al., 2012; Tehranirad et al., 2015) is not computed in the present. This extreme scenario however remains possible as evidenced by the volumes of the deep water deposits identified in Masson et al. (2002) around this archipelago. Nevertheless, we focused on the 80 km$^3$ as it is consistent with the size of the deposits identified at the toe of the volcano, as possibly corresponding to its

last massive flank collapse (about 300,000 years ago). More over, a 500 km$^3$ event would also probably collapse sequentially thus reducing the overall effect (Wynn and Masson, 2003).

The comparison of the different models has proved to be of practical interest in this study, illustrating the complex influence of physical parameters such as dispersion, and numerical ones, such as resolution for instance. It should be encouraged in the future even though is requires a substantial amount of work. For this exercise to be optimized, it is however requested that all the models compared show first convergence of the results which is maybe something lacking in the present study. Nevertheless, convergence is not very easy to demonstrate when variable meshes are used such as in Telemac-2D or SCHISM or even with nested computations such as performed in FUNWAVE-TVD and Calypso. There is certainly some progress to do in this direction in the next years.

## 5  Conclusions

The wave generated by a potential Cumbre Vieja volcano flank collapse and its impact in Europe, and Guadeloupe was studied in this work. The source computation used an improved characterization of the slide rheology compared to previous works. Moreover, the subsequent propagation was performed using different models which allows for a model comparison on a real configuration. The main conclusions of the work performed are the following:

- The new wave source is reduced in half compared to previous estimations mainly due to the larger value of slide viscosity used in this work,

- The wave impact is still very significant on nearby areas, or on more remote coasts, such as Guadeloupe, located on the path of the maximum wave energy for the maximum slide volume considered here (*i.e.*, 80 km$^3$). Smaller slide volumes (*i.e.*, 40 km$^3$ and 20 km$^3$) would have more moderate impacts on these remote areas.

- In Europe, the impact may be considered as moderate to significant in the most exposed areas such as some areas in Portugal and Spain, and weak to moderate along the French Atlantic coast.

- The tsunami source calculated in this paper after 15 minutes of propagation in FUNWAVE-TVD and proposed to the community in the SEANOE repository is dispersive and therefore we recommend to use appropriate models (e.g., Boussinesq models) to propagate further this source in future studies.

- The comparison of the Boussinesq models (*i.e.*, FUNWAVE-TVD and Calypso) mutually validates the models on this particular case and secure the results obtained. With these simulation, we also show that dispersion has a moderate effect on the wave spectrum transformation during propagation.

- Comparisons with NSW models finally illustrated the critical role of the resolution as well as the possible effect of numerical dispersion.

*Data availability.* The new calibrated source (after filtering and propagation in the Boussinesq model) for the La Palma tsunami is made available through the SEANOE portal Abadie et al. (2019).

*Acknowledgements.* This work has been performed in the framework of the PIA RSNR French program TANDEM (Grant no: ANR-11-RSNR-00023-01) and as a part of the project C3AF, funded by the Electricité Réseau Distribution France (ERDF) and the Region Guade-
5   loupe. Part of this work was supported by the Laboratoire de Recherche Conventionné (LRC) CEA-Ecole Normale Supérieure (ENS) Yves Rocard.

**Acronyms**

**ADI** Alternating Direction Implicit.

**EDF** Electricité De France.

**ENS** Ecole Normale Supérieure.

**ERDF** Electricité Réseau Distribution France.

**GMRT** Global Multi Resolution Topography.

**LRC** Laboratoire de Recherche Conventionné.

**MPI** Message Passing Interface.

**NSW** Non-Linear Shallow Water.

**NTHMP** National Tsunami Hazard Mitigation Program.

**SCHISM** Semi-implicit Cross-scale Hydroscience Integrated System Model.

**SEANOE** SEA scieNtific Open data Edition.

**SELFE** Semi-implicit Eulerian-Lagragian Finite-Element.

**SHOM** Service Hydrographique et Océanographique de la Marine.

**TANDEM** Tsunamis in the Atlantic and the English ChaNnel Definition of the Effects through numerical Modeling.

**TVD** Total Variation Diminishing.

**VOF** Volume-Of-Fluid.

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

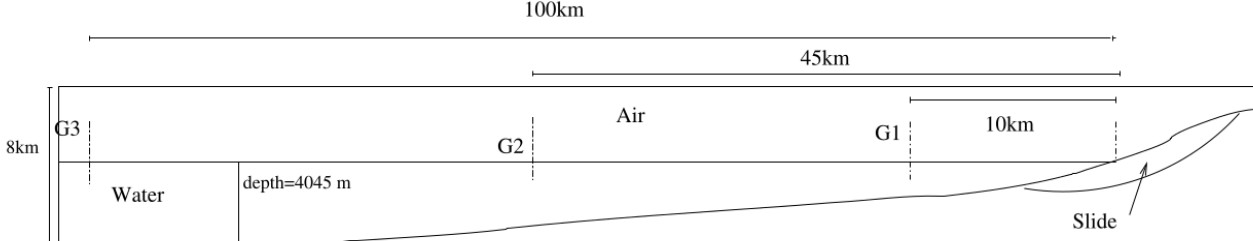

**Figure 1.** Cross section of the 80 km$^3$ La Palma slide scenario considered in Abadie et al. (2012).

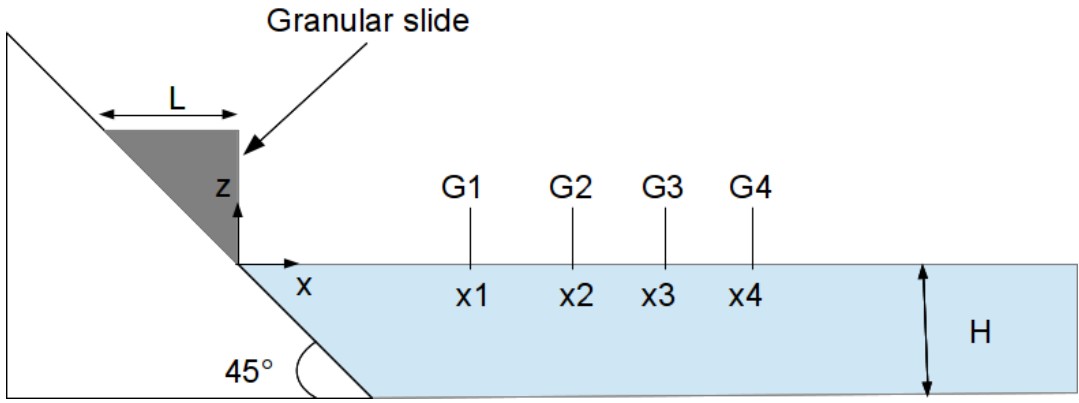

**Figure 2.** Sketch of the experiment performed in Viroulet et al. (2014).

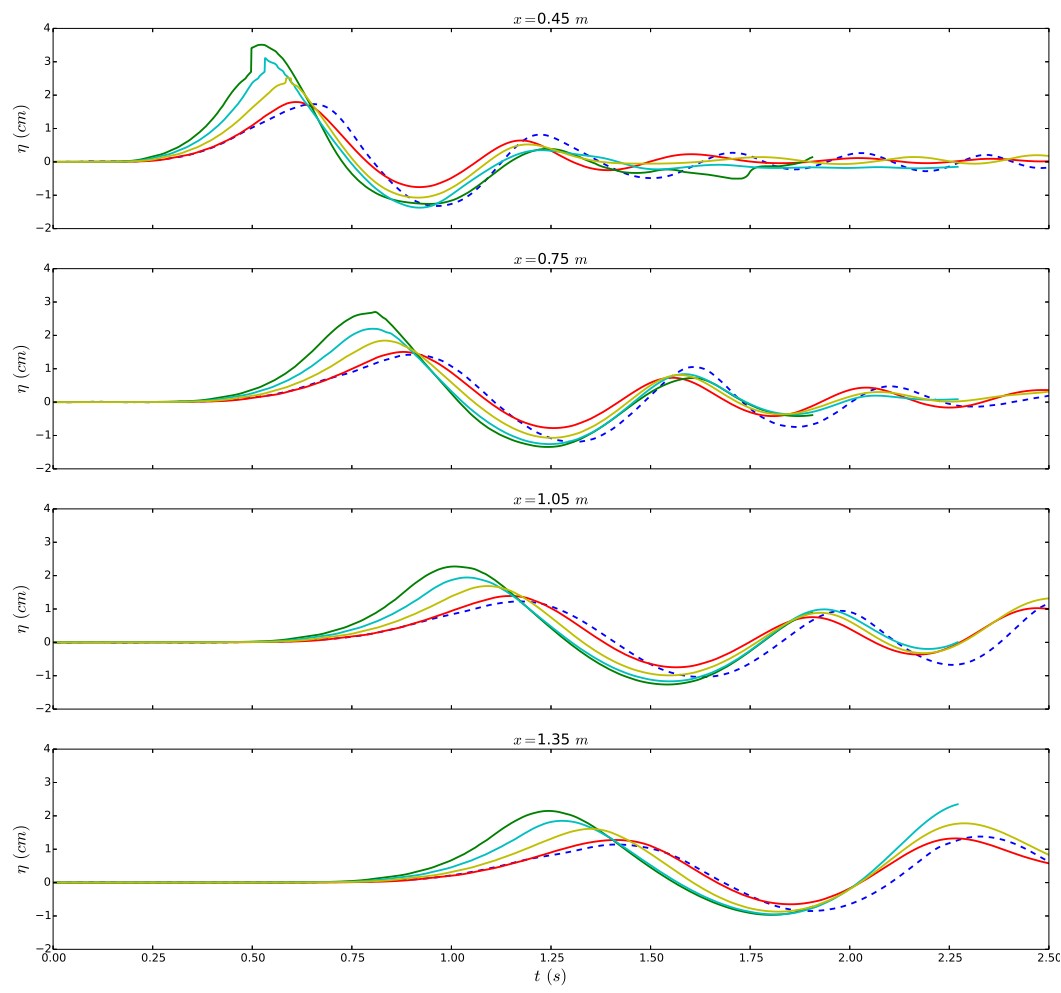

**Figure 3.** Free-surface elevation at the gauges for the experiment (blue dashed line) and the simulations for different values of viscosity, $\mu$=1 Pa·s: green line, $\mu$=2 Pa·s: cyan line, $\mu$=5 Pa·s: yellow line and $\mu$=10 Pa·s: red line.

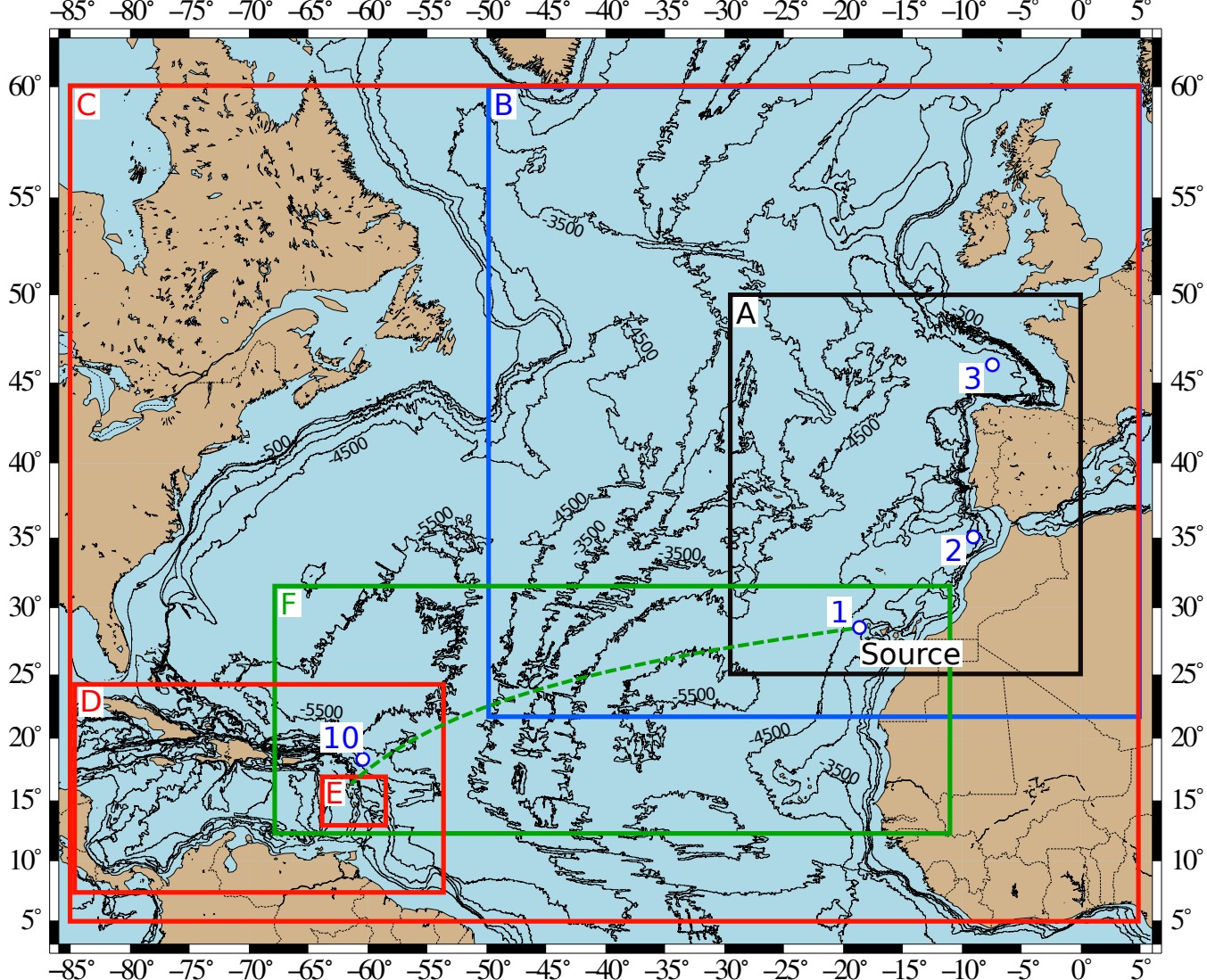

**Figure 4.** Computational domains for Calypso (2-km (A) resolution in black), FUNWAVE-TVD (2.7 km (C), 930 m (D) and 310 m (E) in red), Telemac-2D (in blue (B); variable resolution) and SCHISM (in green (F); variable resolution). The dashed green curve represents the transect between the source and the Guadeloupe Archipelago. Bathymetric contours range from -500 m to -7500 m, every 1000 m. Gauges 1, 2, 3, and 10 are marked by white and blue points.

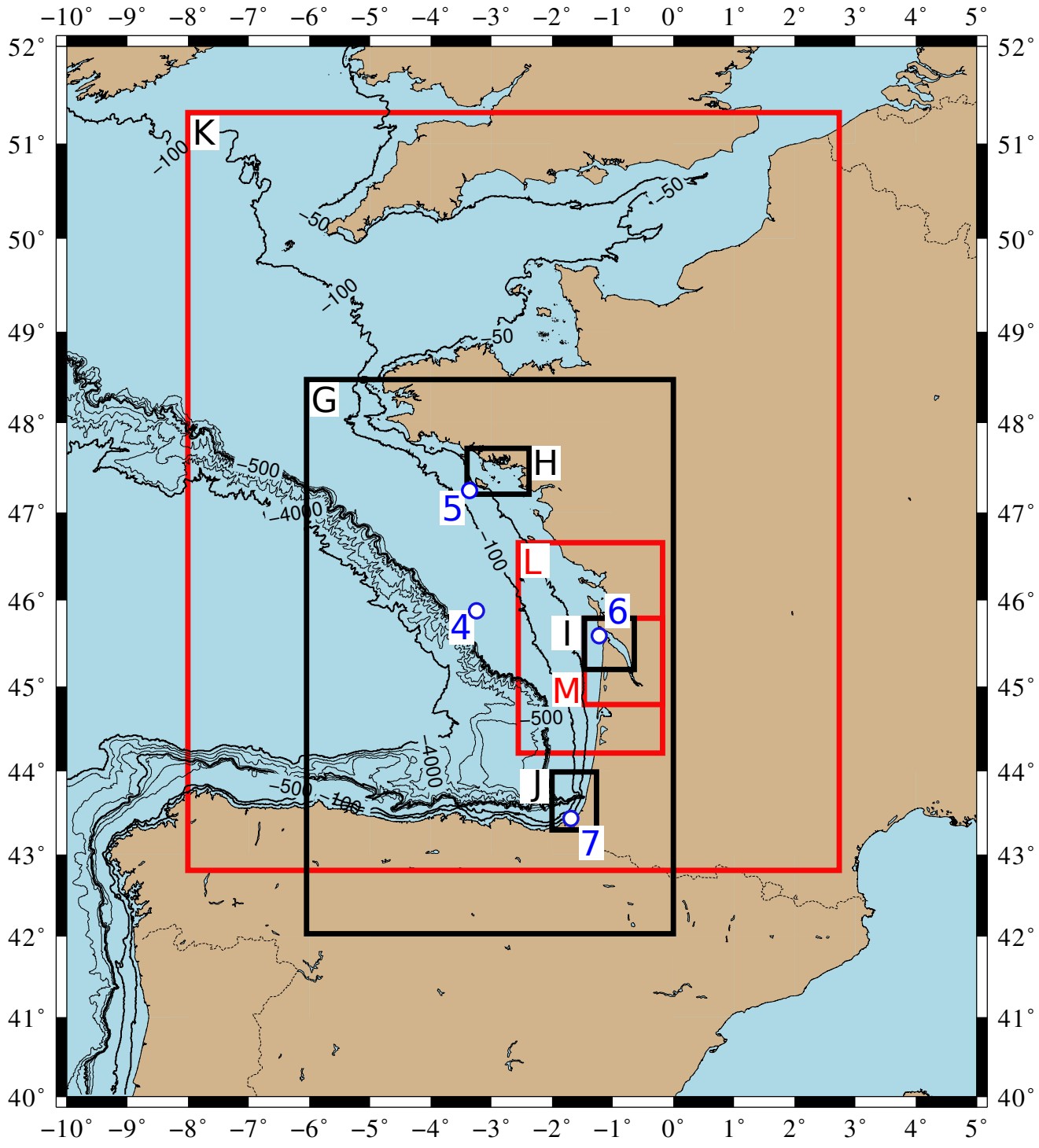

**Figure 5.** Computational domains for Calypso (500-m resolution (G), 125 m (H) or 32.5 m (I and J) in black) and FUNWAVE-TVD (450 m (K), 110 m (L) and 20 m (M) in red). After the -50 m contour, bathymetric contours range from -100 m to -500 m every 100 m, then from -1000 m to -4000 m every 1000 m. Gauges 4, 5, 6 and 7 are marked by the white and blue points.

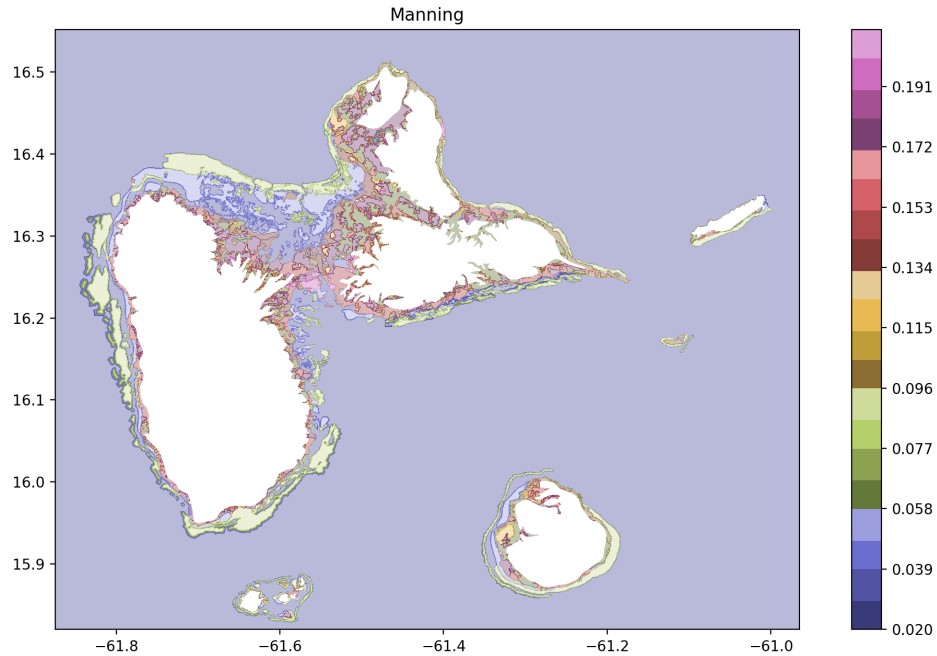

**Figure 6.** Values of Manning coefficient as function of land use in Guadeloupe.

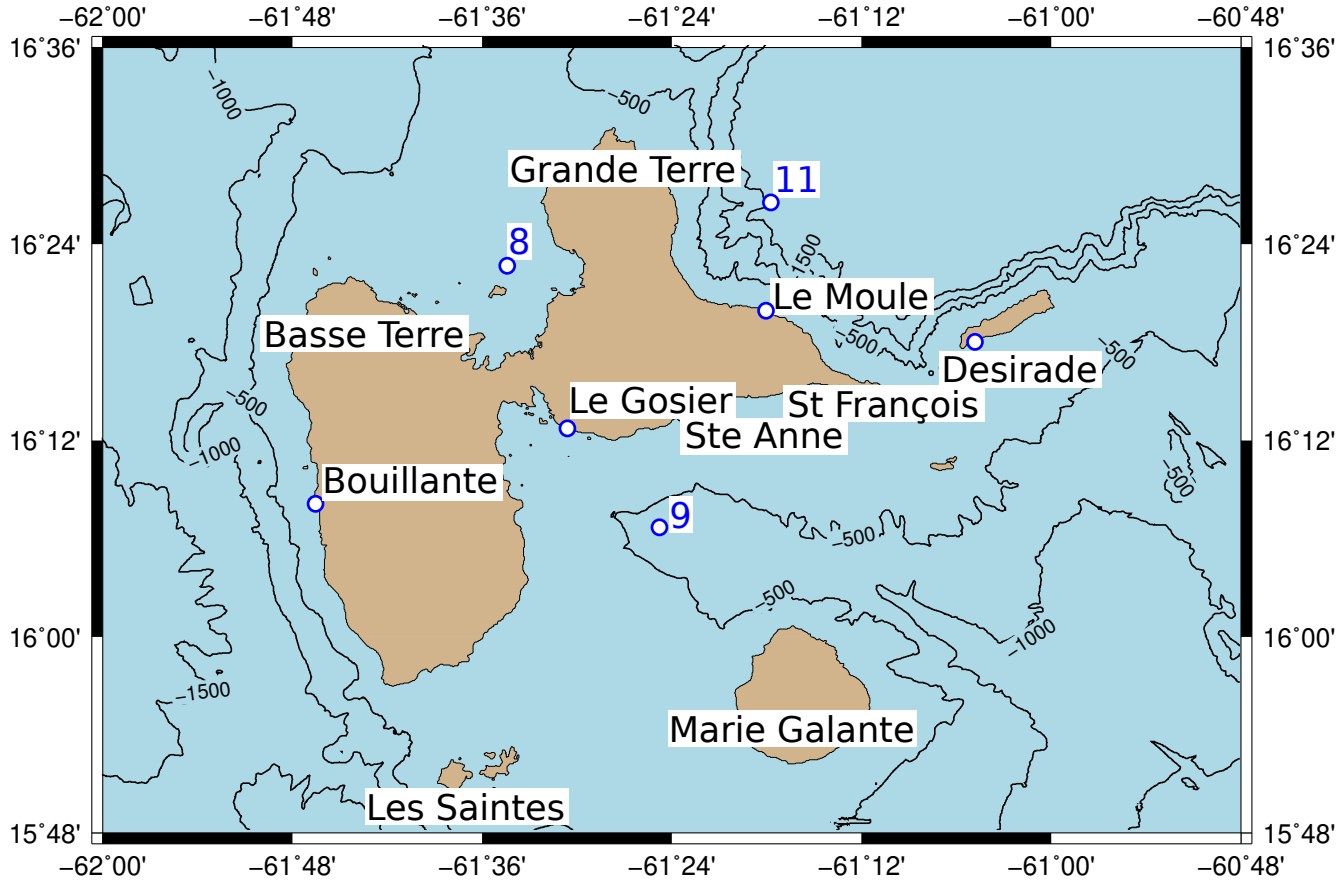

**Figure 7.** Guadeloupe Archipelago and locations of the Gauges 8, 9, and 11 and of the cities of Bouillante, Le Gosier, Le Moule, and Desirade.

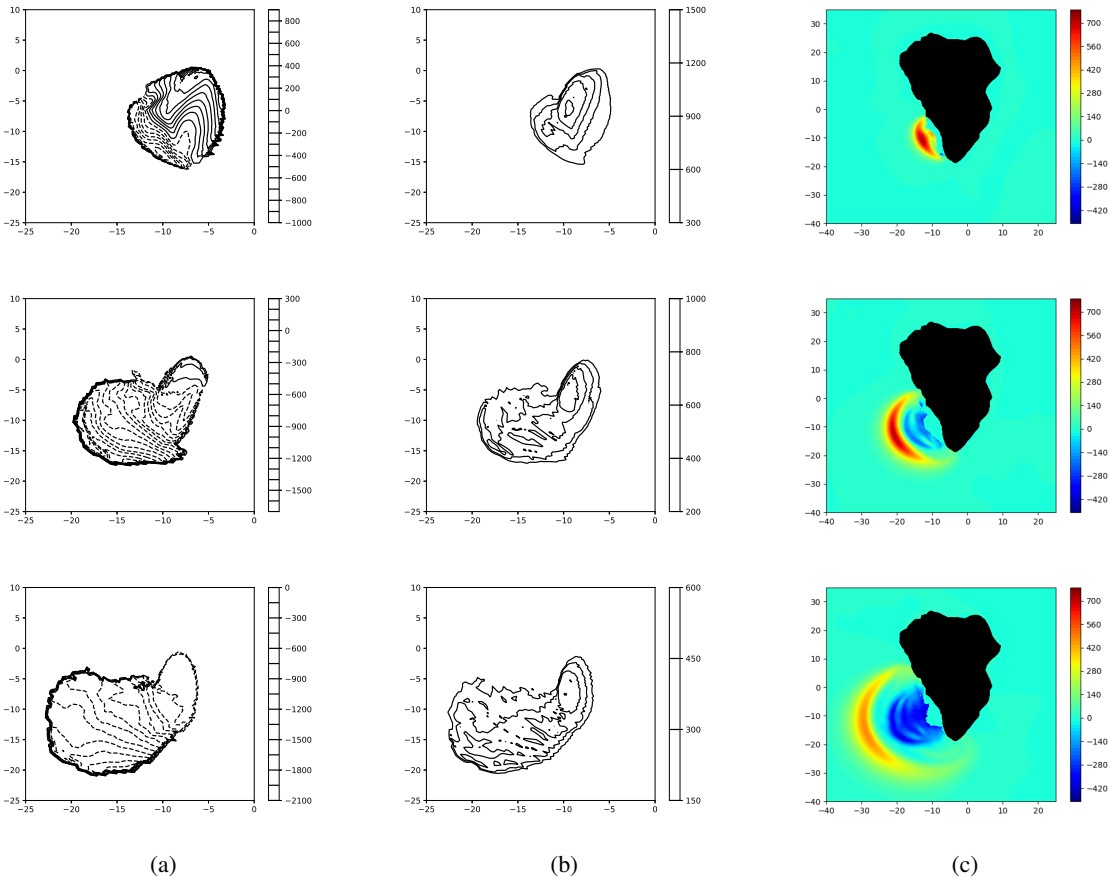

**Figure 8.** Snapshots of slide upper free surface, thickness and corresponding water free surface for the inviscid case (Abadie et al., 2012) ((a), (b), (c)) respectively) for the 80 km³ scenario at t=60 s (row 1), 120 s (row 2), 180 s (row 3).

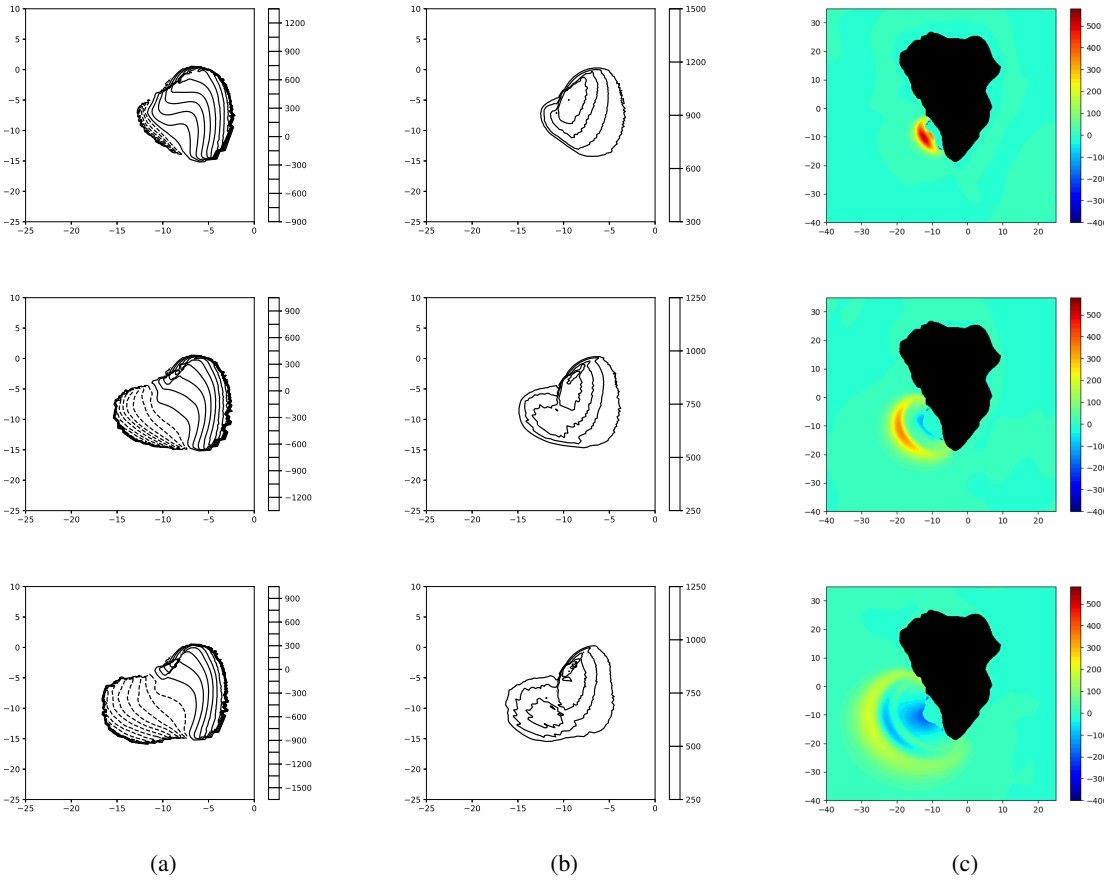

(a)                        (b)                        (c)

**Figure 9.** Snapshots of slide upper free surface, thickness and corresponding water free surface for the present study (*i.e.*, viscous slide with a viscosity of $2\times10^7$ Pa·s, ((a), (b), (c)) respectively) for the 80 km$^3$ scenario at t=60 s (row 1), 120 s (row 2), 180 s (row 3).

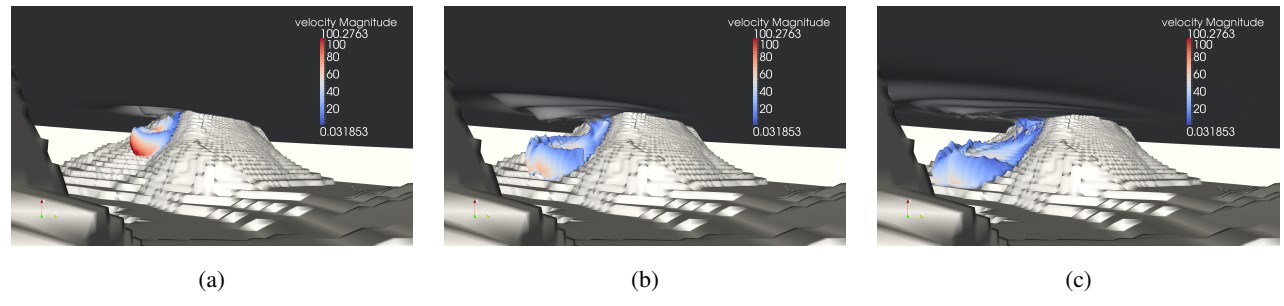

|     |     |     |
| :-: | :-: | :-: |
| (a) | (b) | (c) |

**Figure 10.** THETIS 3D computations for 80 km$^3$ slide volume. Snapshots of 0.1 slide volume fraction contour colored by velocity magnitude, at $(a)$: t=100 s, $(b)$: 200 s and $(c)$: 300 s. Inviscid slide (Abadie et al., 2012).

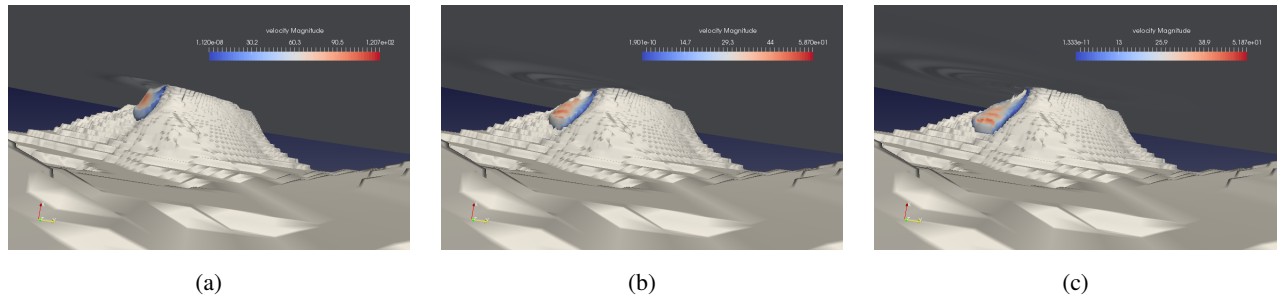

(a)                                              (b)                                              (c)

**Figure 11.** THETIS 3D computations for 80 km$^3$ slide volume. Snapshots of 0.1 slide volume fraction contour colored by velocity magnitude, at $(a)$: t=102 s, $(b)$: 230 s and $(c)$: 342 s. Slide viscosity $2 \times 10^7$ Pa·s.

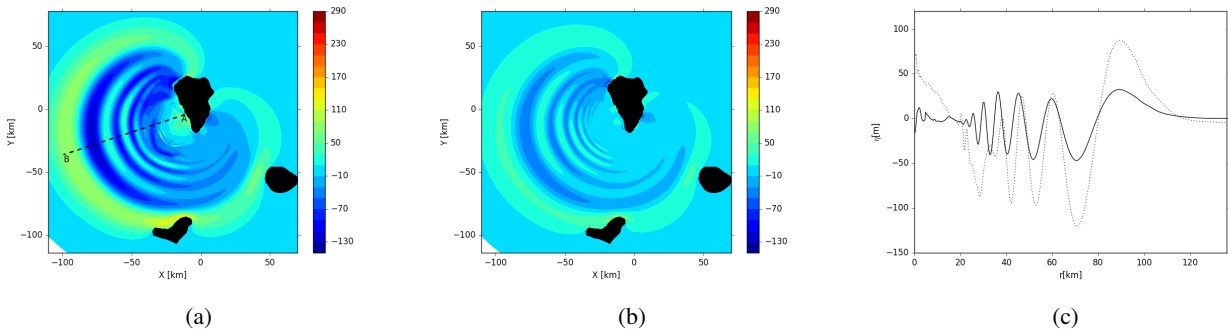

(a)                (b)                (c)

**Figure 12.** THETIS 3D computations for the 80 km$^3$ slide scenario, $t \approx 560$ s. $(a)$: Inviscid slide. $(b)$: Slide viscosity $2 \times 10^7$ Pa·s. $(c)$: Free surface elevations along the cross section A-B of Frame (a).

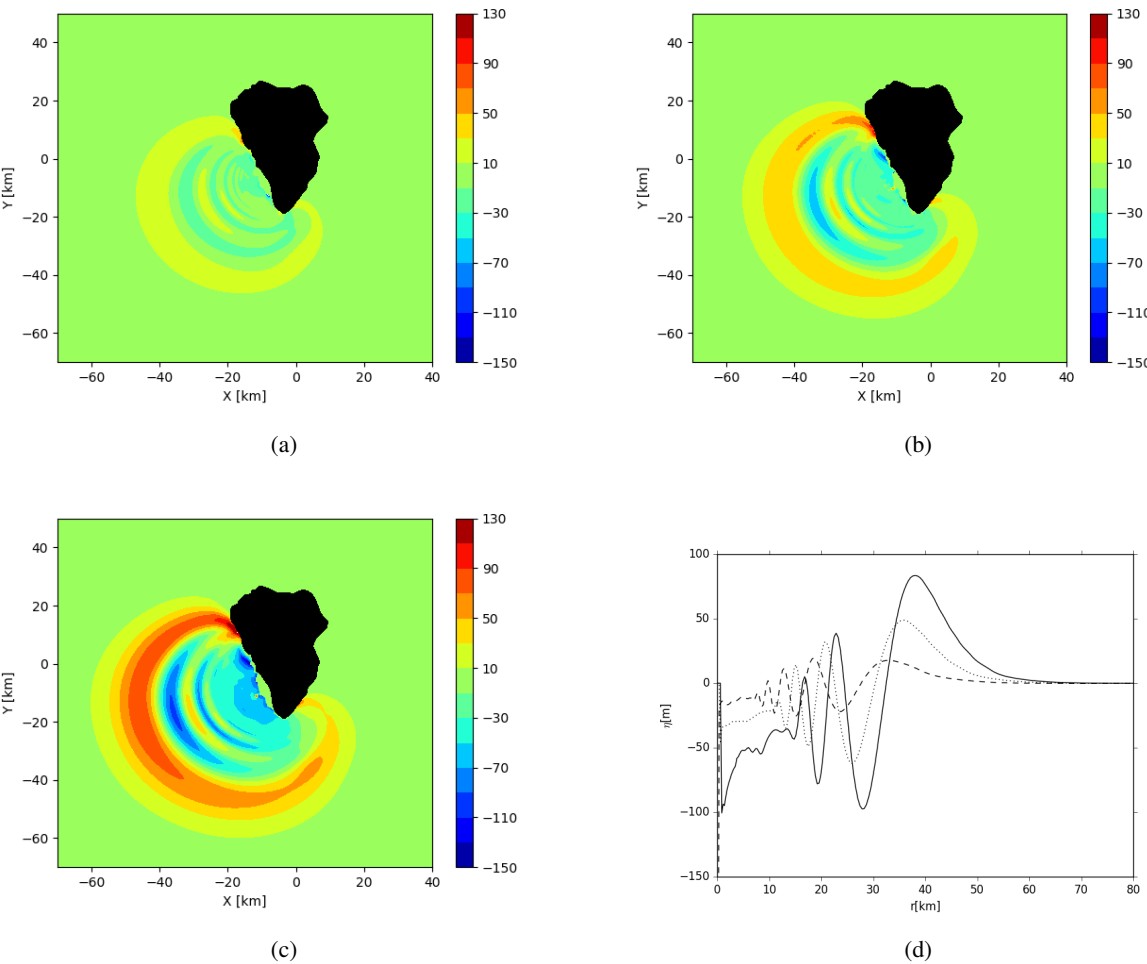

**Figure 13.** THETIS 3D computations for $(a)$: 20 km$^3$, $(b)$: 40 km$^3$ and $(c)$: 80 km$^3$ slide scenarios at t=5 min. $(d)$: Free surface elevations along section A-B of Frame (a) of Figure 12. Slide viscosity $2 \times 10^7$ Pa·s.

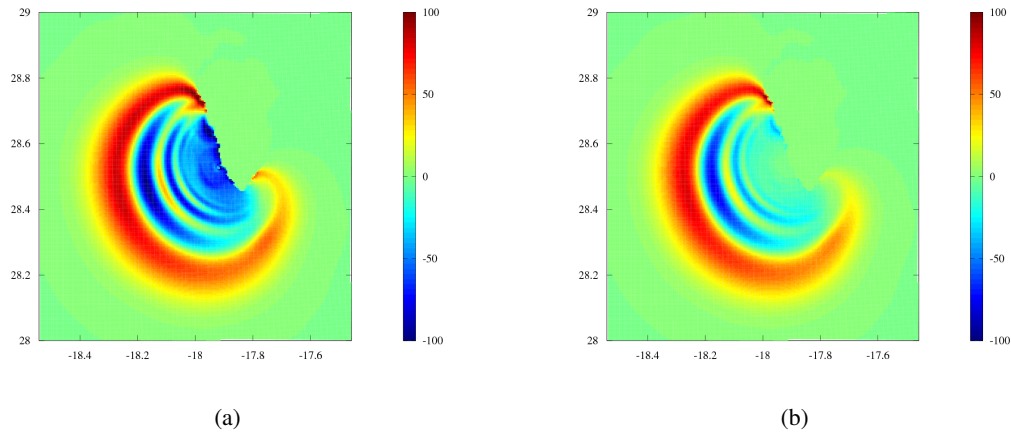

(a)                                                                (b)

**Figure 14.** Region around Cumbre Vieja Volcano after 5 minutes of simulated time with THETIS for the 80 km$^3$ slide scenario. ($a$): Wave elevation for the initial solution from THETIS. ($b$): Filtered state which is used to initialize FUNWAVE-TVD.

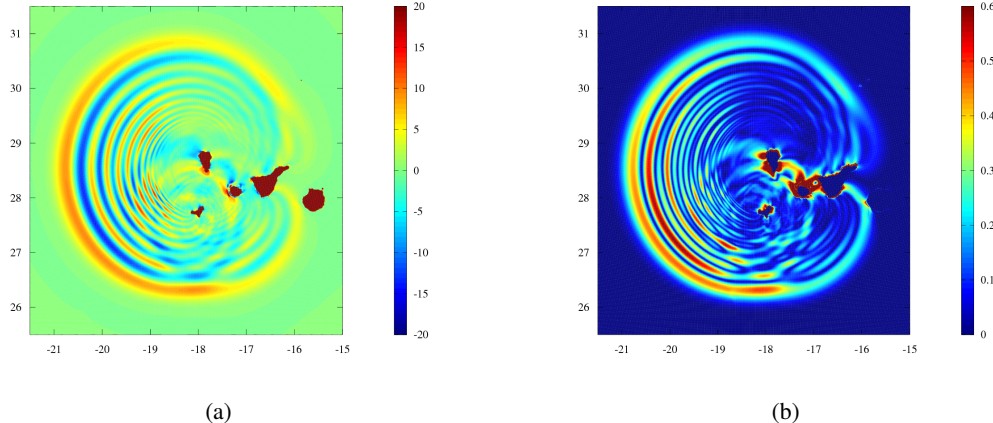

(a)                                                    (b)

**Figure 15.** Region around the Canary Islands, 20 minutes after the beginning of the event (after 5 minutes of simulated time with THETIS, and 15 minutes of simulated time with FUNWAVE-TVD) during the 80 km$^3$ slide volume scenario. ($a$): Wave elevation. ($b$): Horizontal water velocity magnitude.

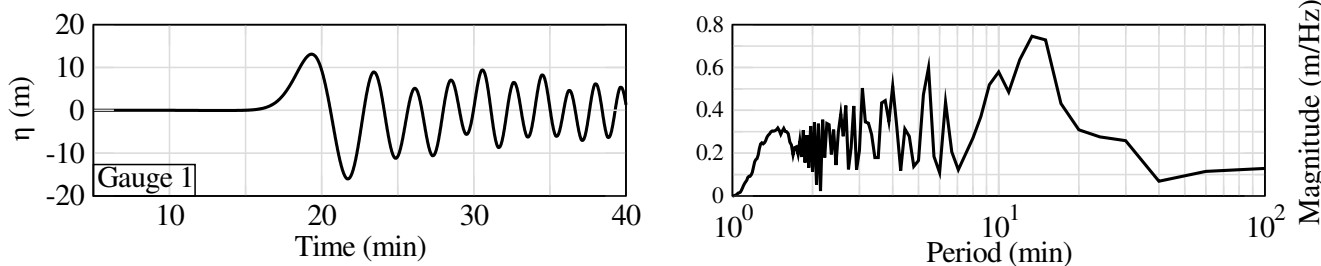

**Figure 16.** Surface elevation (m, left) and associated Fourier transform (right) for the 80 km$^3$ scenario at Gauge 1 close to the source. The time takes into account the 20 first minutes of the slide and tsunami generation.

## At the North-Atlantic scale

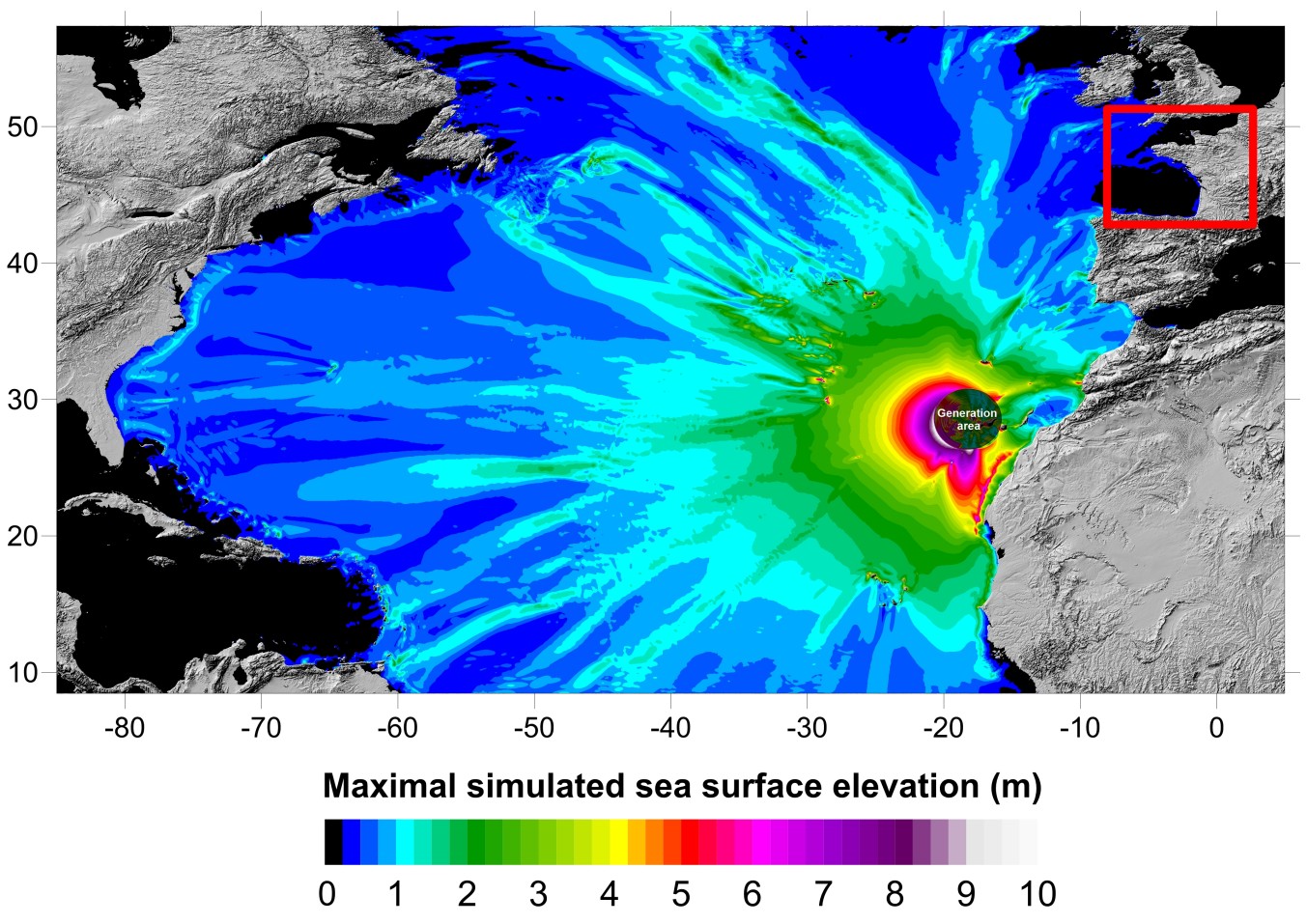

**Maximal simulated sea surface elevation (m)**

0  1  2  3  4  5  6  7  8  9  10

**Figure 17.** Maximum surface elevations (m) computed by FUNWAVE-TVD for the 80 km$^3$ scenario, from the generation area to the French coasts and other remote territories, with a 2.7-km resolution. The red rectangle represents a daughter grid covering the western French coasts (see Figures 5 and 18).

# Near metropolitan France

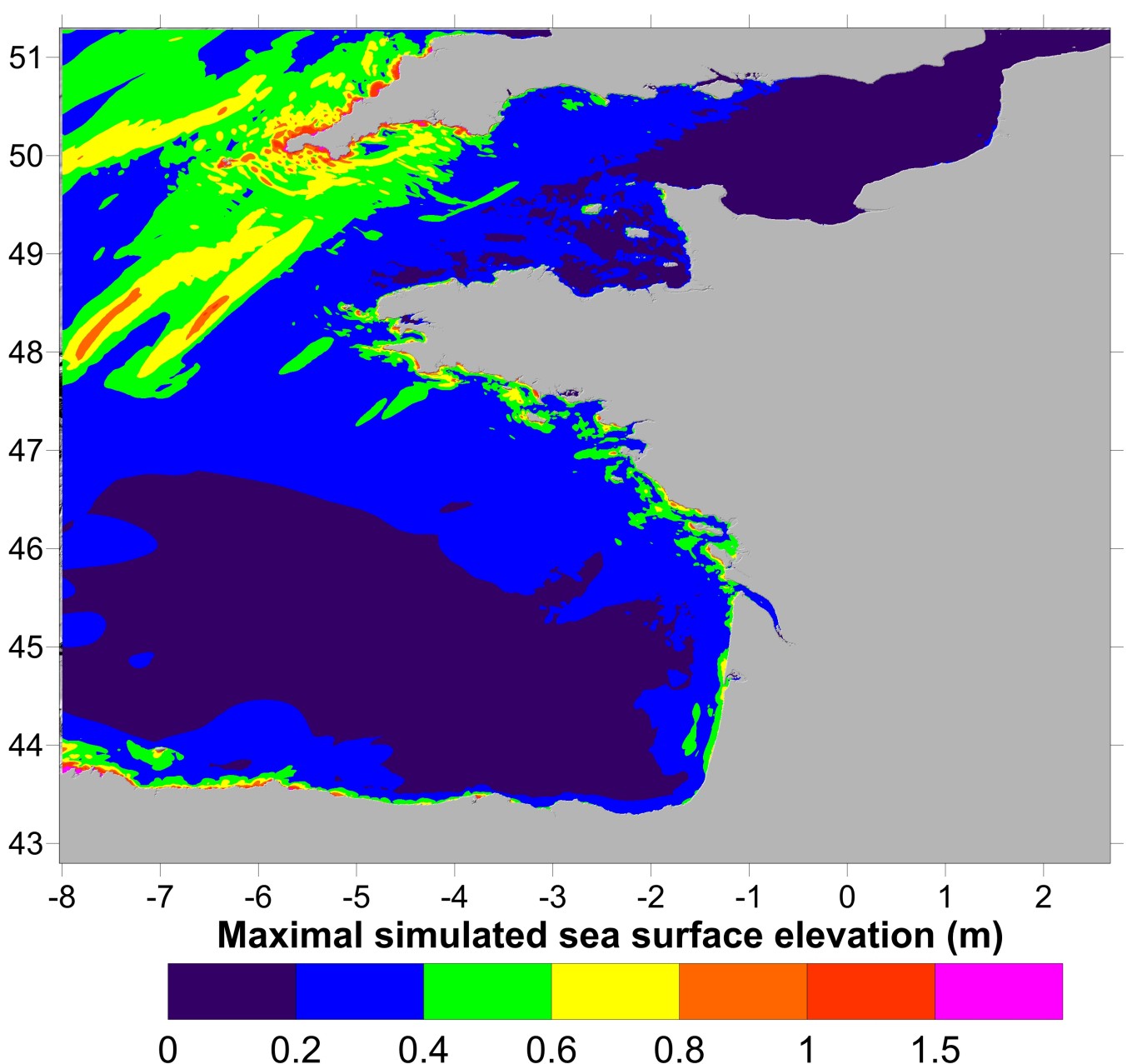

**Maximal simulated sea surface elevation (m)**

**Figure 18.** Maximum surface elevations (m) computed by FUNWAVE-TVD for the 80 km$^3$ scenario on the western French coasts, with a 450-m resolution.

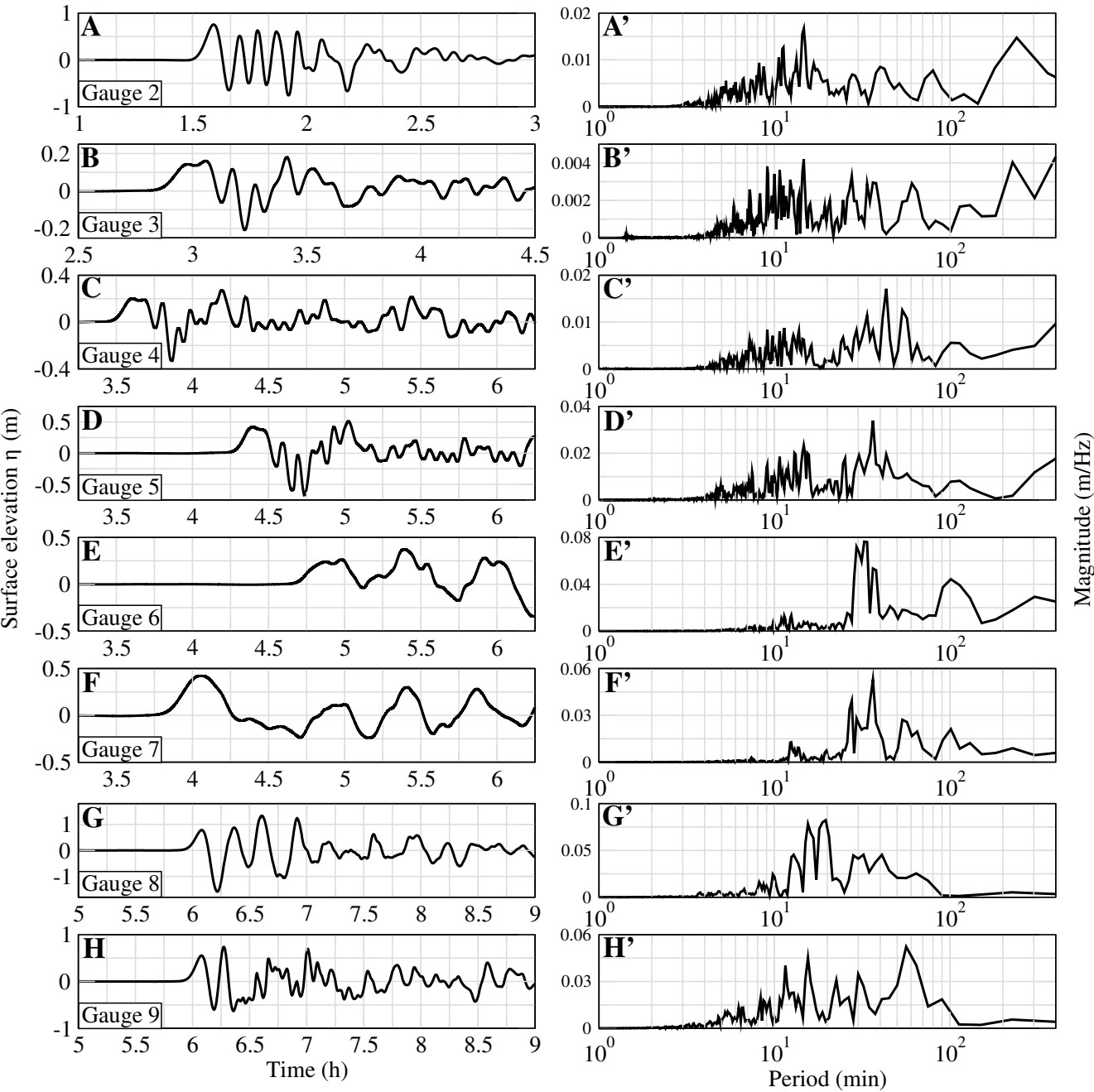

**Figure 19.** Surface elevations (m) (left column) and Fourier transforms (right column) for the 80 km$^3$ scenario at Gauge 2 in south Portugal (A and A'), Gauge 3 in the abyssal plain of the Bay of Biscay (B and B'), Gauge 4 in the continental shelf of the Bay of Biscay (C and C'), Gauge 5 in south Brittany (D and D'), Gauge 6 in the Gironde estuary (E and E'), Gauge 7 in Saint-Jean-de-Luz (F and F') and Gauges 8 and 9, respectively north (G and G') and south (H and H') of Guadeloupe, computed by FUNWAVE-TVD. The time takes into account the 20 first minutes of the slide and tsunami generation.

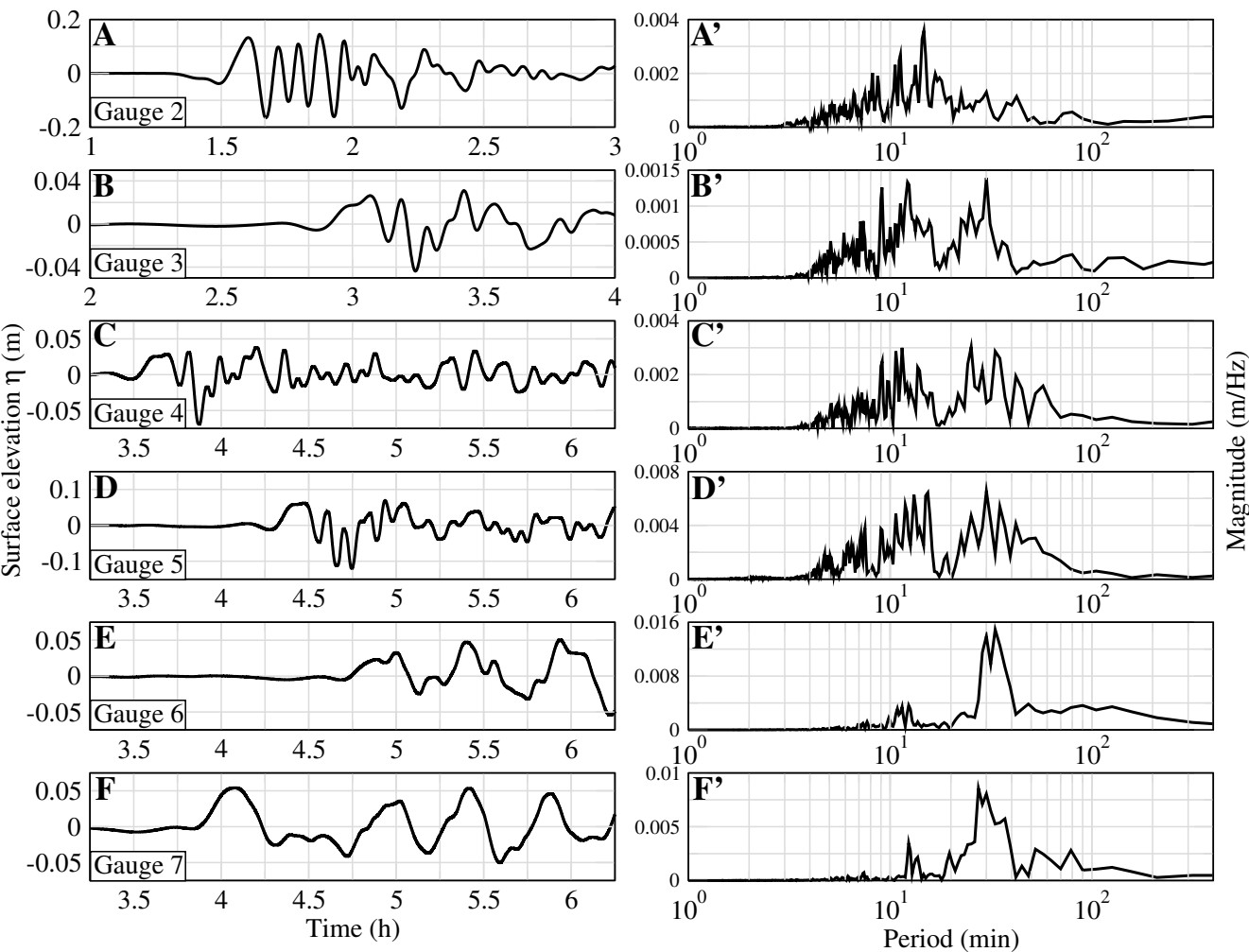

**Figure 20.** Surface elevations (m) (left column) and Fourier transforms (right column) for the 20 km$^3$ scenario at Gauge 2 in south Portugal (A and A'), Gauge 3 in the abyssal plain of the Bay of Biscay (B and B'), Gauge 4 in the continental shelf of the Bay of Biscay (C and C'), Gauge 5 in south Brittany (D and D'), Gauge 6 in the Gironde estuary (E and E') and Gauge 7 in Saint-Jean-de-Luz (F and F'), computed by FUNWAVE-TVD. The time takes into account the 20 first minutes of the slide and tsunami generation.

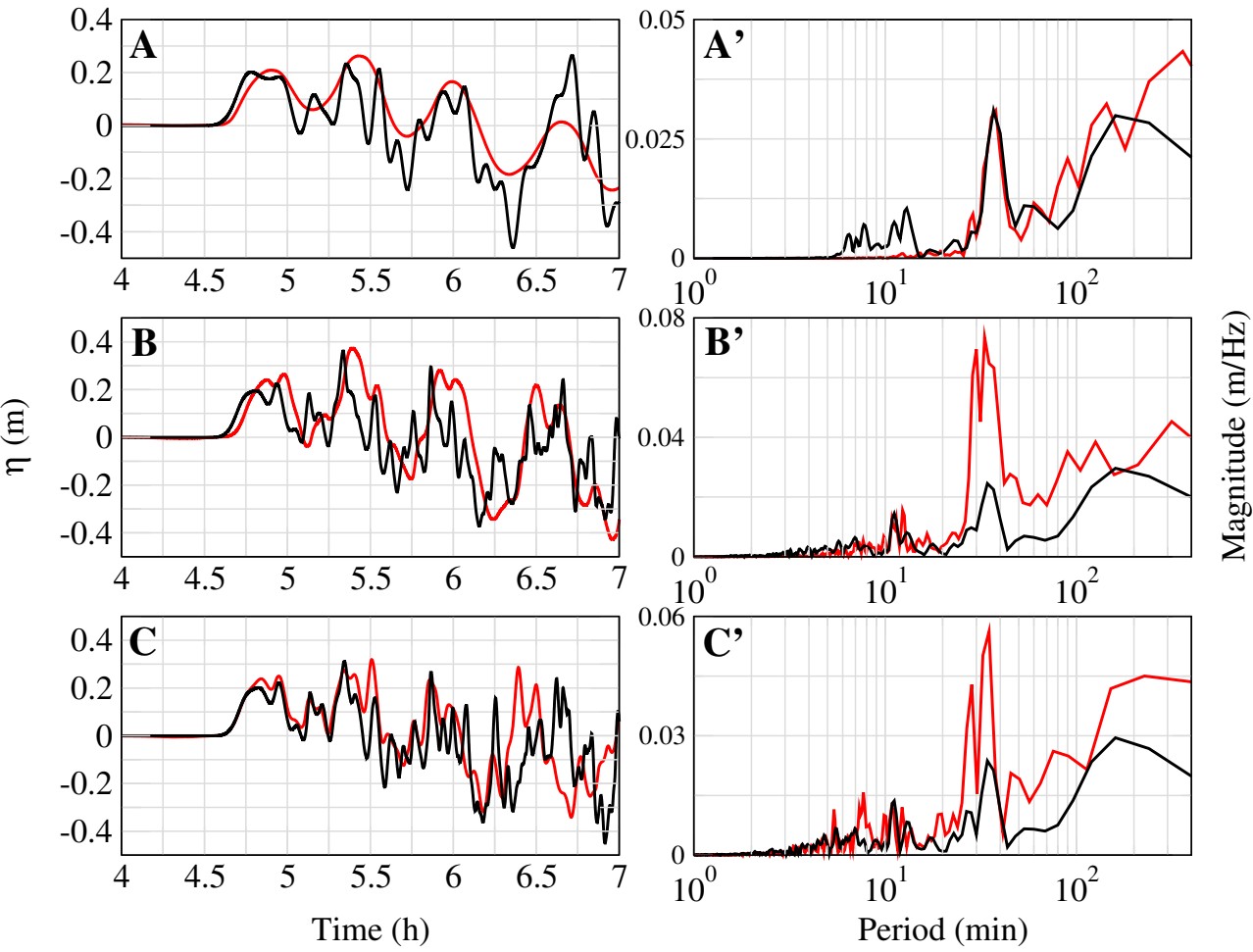

**Figure 21.** Comparison of the surface elevation (m) and the associated periods computed by a Fourier transformation (A', B' and C') at Gauge 6 between Calypso (in black) and FUNWAVE-TVD (in red) for the 80 km$^3$ scenario at the Gironde estuary, for three resolutions: 2.7 km (A), 450 m (B) and 110 m (C) for FUNWAVE-TVD and 2 km (A), 500 m (B) and 125 m (C) for Calypso. The time takes into account the 20 first minutes of the slide and tsunami generation.

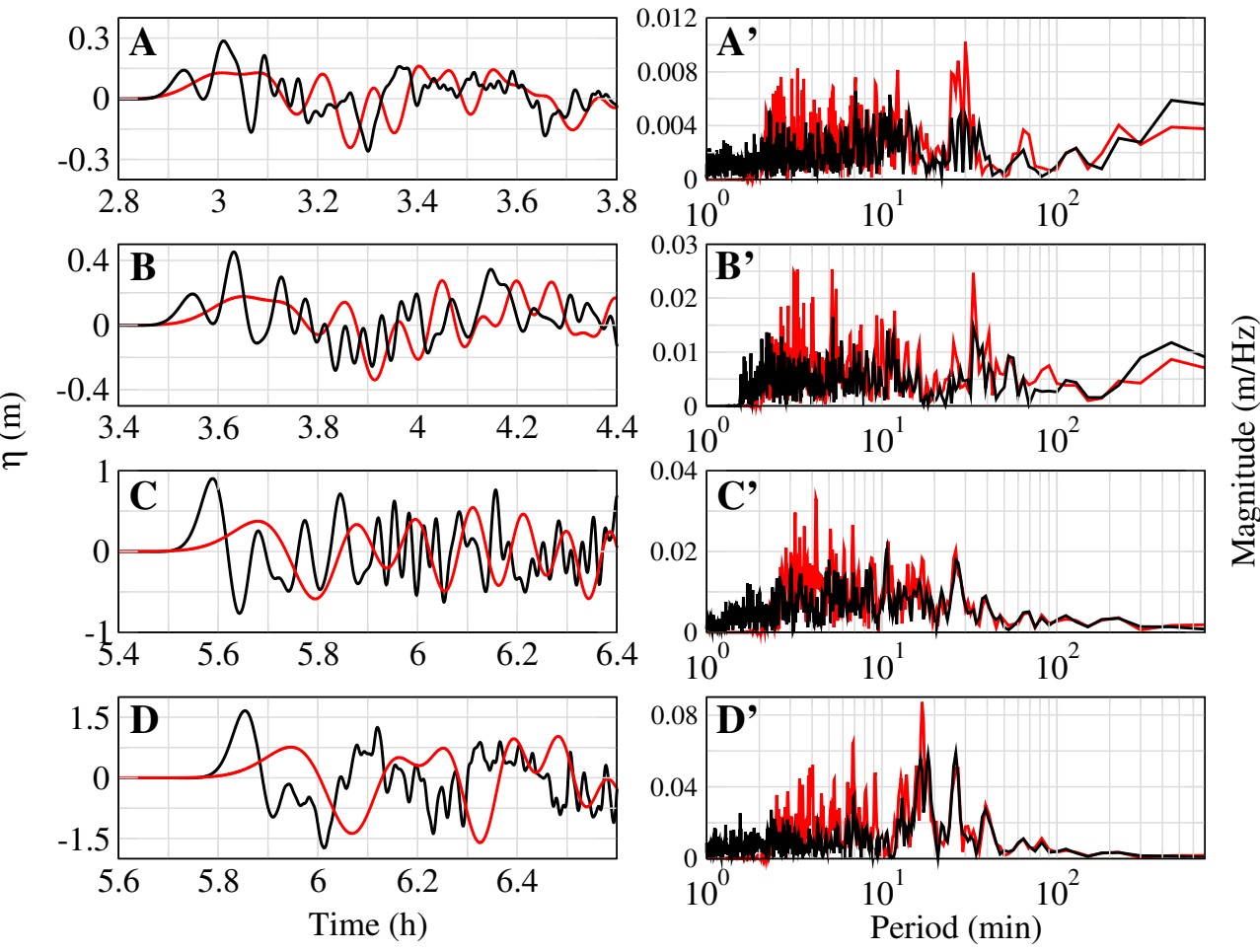

**Figure 22.** Comparison of the surface elevation (m) computed by NSW (in black) and Boussinesq (in red) models of Calypso with resolution 2 km at Gauges 3 (A), 4 (B), 10 (C) and 11 (D) for the 80 km$^3$ scenario and the associated periods computed by a Fourier transformation (A', B', C' and D'). The time takes into account the 20 first minutes of the slide and tsunami generation.

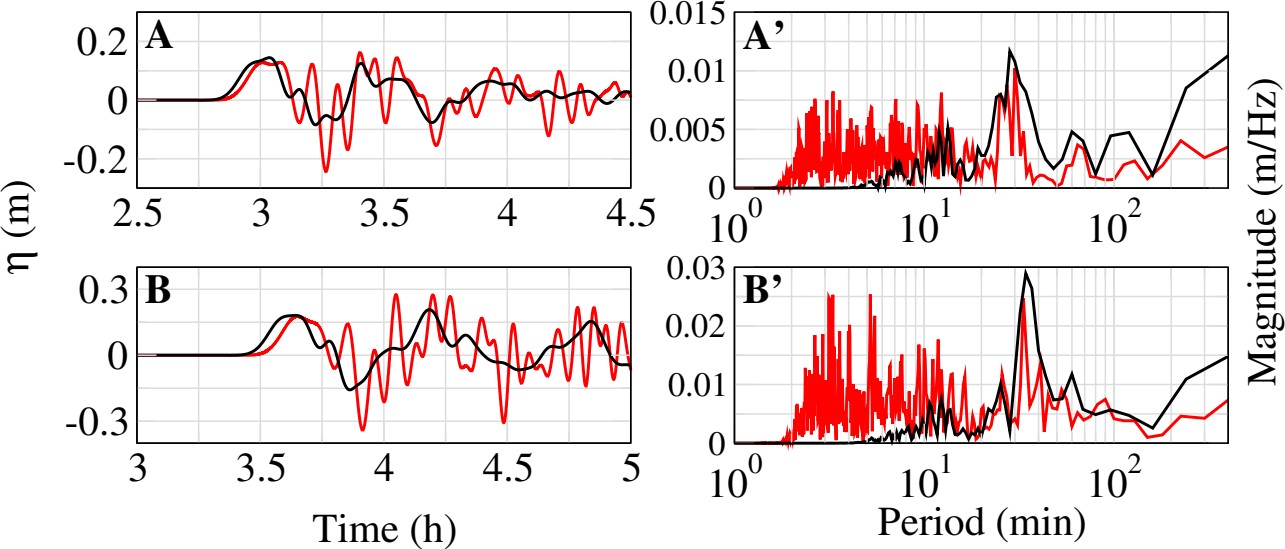

**Figure 23.** Comparison of the surface elevation (m) computed by the Boussinesq model of Calypso (in black) with resolution 2 km and Telemac-2D (in red) at Gauges 3 (A) and 4 (C) for the 80 km³ scenario and the associated periods computed by a Fourier transformation (A', B', C' and D'). Panels B and D show the averaged results of Calypso (in black), still compared to Telemac-2D (in red). The time takes into account the 20 first minutes of the slide and tsunami generation.

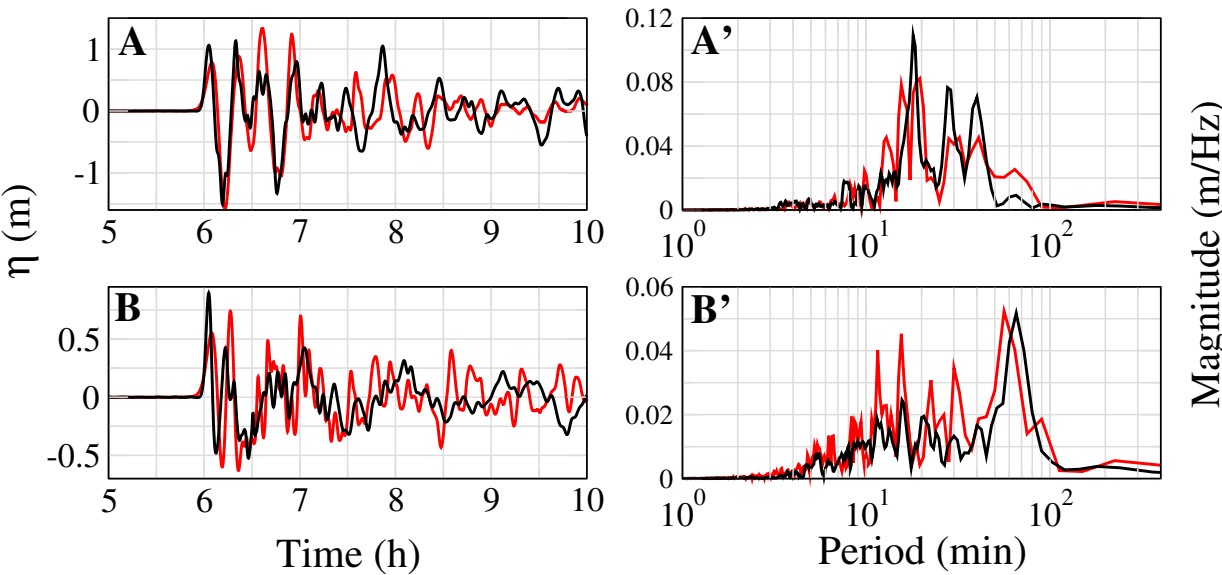

**Figure 24.** Comparison of the surface elevation (m) and the associated periods computed by a Fourier transformation between SCHISM (black) and FUNWAVE-TVD (red) in the vicinity of the Guadeloupe area, at the Gauges 8 (A and A') and 9 (B and B') mentioned on Figure 7 for the 80 km$^3$ scenario. The time takes into account the 20 first minutes of the slide and tsunami generation.

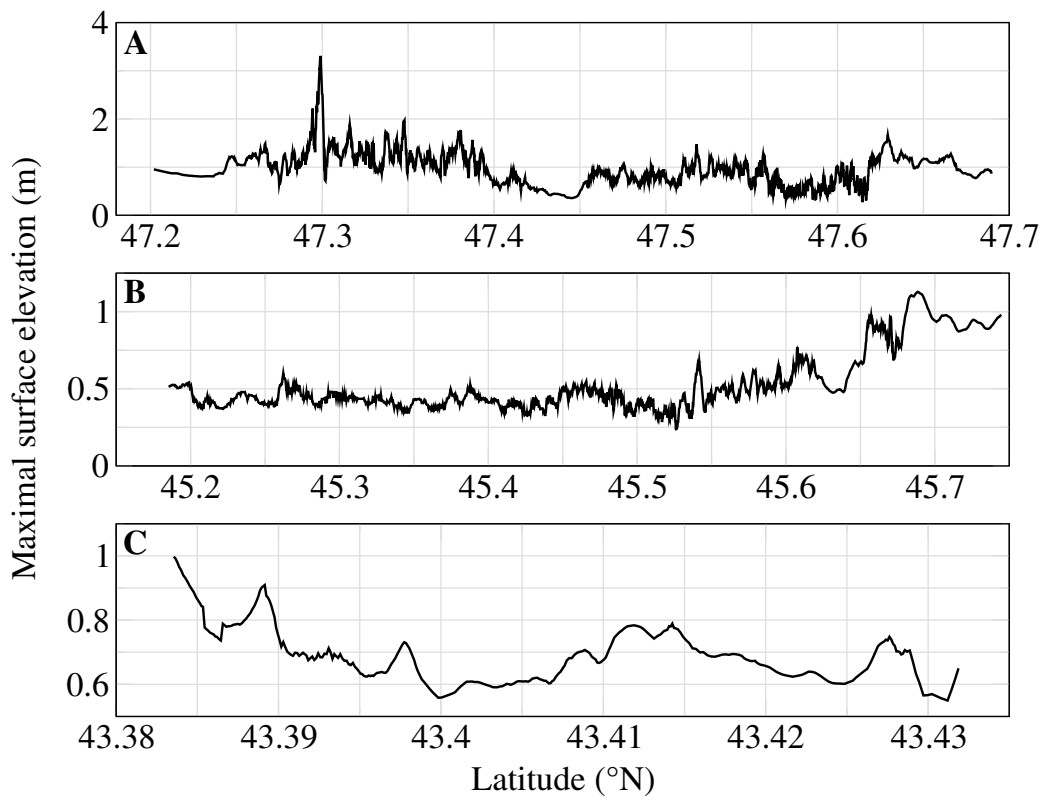

**Figure 25.** Maximum surface elevation computed with Calypso-B-NSW for the 80 km$^3$ scenario using the finest grid resolution at isobath -5 m for different areas along the French Atlantic coastline. Panel A: Morbihan, panel B: Gironde estuary, panel C: Saint-Jean-de-Luz area

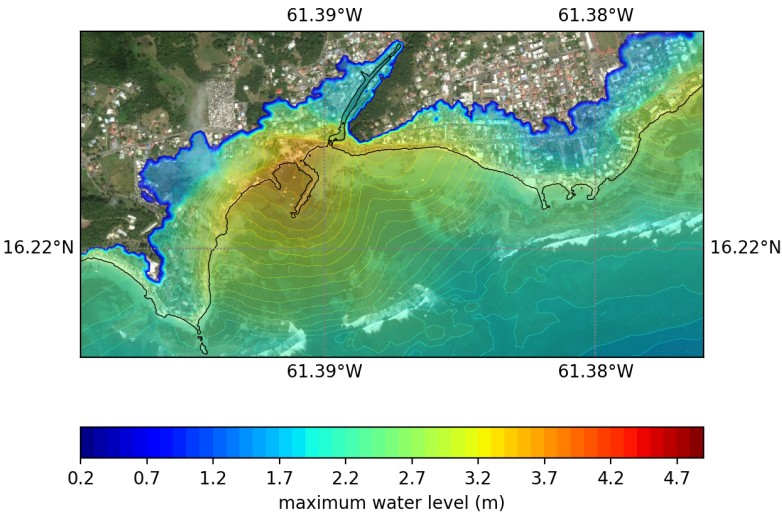

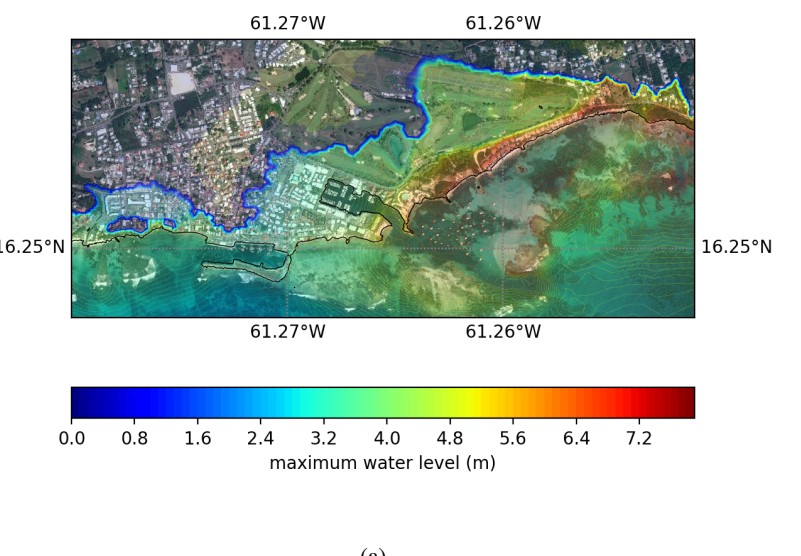

(a)                                                                                          (b)

**Figure 26.** Flood map showing the maximum water level reached during the 80 km$^3$ scenario for the region of Sainte-Anne (a) and Saint-François (b) (see locations in Figure 7), computed by SCHISM. Map created using ArcGIS® software by Esri. ArcGIS® and ArcMap™ are the intellectual property of Esri and are used herein under license. Copyright © Esri. All rights reserved.

**Table 1.** Summary of locations of numerical output (see Figure 4 for Gauges 1, 2, 3, and 10, Figure 5 for Gauges 4, 5, 6, and 7, and Figure 7 for Gauges 8, 9, and 11).

| Gauge | Latitude | Longitude | Depth (m) |
|-------|----------|-----------|-----------|
| 1 | 27.7 | -19.8 | 4430 |
| 2 | 35.2247 | -8.85923 | 3260 |
| 3 | 45.8663 | -6.85191 | 4810 |
| 4 | 46.0016 | -3.27661 | 130 |
| 5 | 47.2934 | -3.26421 | 50 |
| 6 | 45.5854 | -1.21069 | 10 |
| 7 | 43.3979 | -1.67607 | 20 |
| 8 | 16.379519 | -61.582708 | 110 |
| 9 | 16.1 | -61.41 | 620 |
| 10 | 17.9758 | -60.0239 | 6500 |
| 11 | 16.4454 | -61.3164 | 1200 |

**Table 2.** Summary of grid characteristics (see Figure 4 for footprints of grids A, B, C, D, E, and F, and Figure 5 for footprints of grifs G, H, I, J, K, L, and M)

| Grid | Code | Resolution |
|------|------|------------|
| A | Calypso | 2 km |
| B | Telemac-2D | variable |
| C | FUNWAVE-TVD | 2.7 km |
| D | FUNWAVE-TVD | 930 m |
| E | FUNWAVE-TVD | 310 m |
| F | SCHISM | variable |
| G | Calypso | 500 m |
| H | Calypso | 125 m |
| I | Calypso | 32.5 m |
| J | Calypso | 32.5 m |
| K | FUNWAVE-TVD | 450 m |
| L | FUNWAVE-TVD | 110 m |
| M | FUNWAVE-TVD | 20 m |