# Peer review of "La Palma landslide tsunami: computation of the tsunami source with a calibrated multi-fluid Navier-Stokes model, hazard assessment, and model intercomparison."

_Natural Hazards and Earth System Sciences, 2019_

## Referee Comment (RC1) · Anonymous Referee #1 · 22 Aug 2019

This paper reports on important information for tsunami hazard assessment for countries with exposed costs located or connected to the Atlantic Ocean Basin. As stated by the authors, also a Cumbre Vieja Volcanic (CVV) collapse represents an extreme source of likely very low probability, it must nevertheless be considered in a comprehensive tsunami hazard assessment study. For this reason, this paper eventually warrants publication in a journal such as NHESS, after some of the concerns detailed below are addressed.

Review comments

[Figure]

In introduction, the paper lists some earlier work on CVV tsunami modeling and far-field impact but does not really quantitatively summarize all the important findings. Some references also are missing, for instance Grilli et al. (2016).

In introduction, the paper should better discuss why various propagation models are used to simulate the same cases (for instance to assess epistemic uncertainty). It seems that models were used only based on their availability or the experience within a certain group and not because they really featured the necessary physics. As early as Løvholt et al. (2008), it was made clear that CVV collapses generate highly dispersive wavetrains. This was also reported in the far-field by Tehranirad et al. (2015). Here, both dispersive and non-dispersive models are used and although some of the results presented (particularly in and around Guadeloupe) clearly show significant dispersive effects that in a standard way reduce the leading wave elevation and enhance that of the second and/or third waves and then shed a larger tail of oscillation, the conclusion is more or less that all models are acceptable to simulate this event, as this were a forgone conclusion ?

Also, the conclusions wrongly imply that dispersive effects only express themselves early in the event, during the THETIS and FUNWAVE near-field simulations which are both dispersive, and hence it is fine using a non-dispersive model for the transoceanic propagation. This contrary to established effects of dispersion, which a train of short enough waves, will go growing with time and distance. See for instance the discussion and dispersion indicators (which could be computed here) detailed in Glimsdal et al. (2013), and also applied in Schambach et al (2019) for landslide tsunamis. In this reviewer's opinion, the results presented here would lead to conclude using TELEMAC-2D or SCHISM for anything other then the nearshore inundation should be ruled out. See for instance Grilli et al. (2016) for which the transoceanic propagation of the CVV tsunami was done with FUNWAVE and the local impact on Hispaniola was computed with TELEMAC-2D in an unstructured grid.

Here the authors present new, more realistic, CVV sources, but the comparison between the old sources and the new ones is quite limited. More details would be desirable for both the wave generation and the slide. To begin with, the initial topography and geometry used for each source (both old and new ones and all the considered volumes), such as shown in Fig. 1, should be better described (is it fully available on some data storage site ?). Then, it would be desirable to clearly show stages of the slide development underwater, for the old inviscid source and the new ones. In this respect, Figs. 8 and 9 are hard to get any information from. Having both comparative cross section and planview of slide flow and runout as a function of time would be desirable. For the generated tsunami, only Fig. 10 shows some measure of comparison but there as well more comparison allowing the reader to understand the nature of the tsunami source and how it is affected by viscosity would be helpful. P5 l8, it is mentioned the landslide continues to move for a very long time as in Abadie et al (2012) but this should be illustrated in a figure.

There is no real review of other models that have been used for simulating subaerial or submarine landslide tsunamis, for dense fluid or granular mediums. I suggest for instance to refer to and at least briefly discuss models of Horrillo et al. (2013), Ma et al. (2015) and Kirby et al. (2016) and Løvholt et al. (2017). See also the recent application of the Ma et al. and Kirby et al. models to a volcanic collapse tsunami in Grilli et al. (2019).

Model grids are not clearly enough defined. Better (more readable, less dark) figures of grids on a global map, plus some regional zoom outs, and tables of parameters are required. Lines 22-29 p 5 are too implicit. For instance, for SCHISM ii t is said that "the resolution is adapted to accurately reproduce wavetrains of 2 minutes or more with at least 20 nodes per wavelength in deep ocean". Is this applicable to all 4 models ? how was this decided about ?

Please explicitly explain models and grid used and how. L32 p5, saying all models use the same basic equations is meaningless. Please specify common physical assumptions and differences among models. Listing eqs. for 3 out of 4 of the models without

background is meaningless. Equations are not necessary here but model physics, features and limitations are. If the authors do want to list eqs., they should do it for all models and in appendix. The end of section 2.3 would be a good discuss anticipated dispersive effects based on earlier work on landslide tsunamis and on CVV.

Regarding Calypso, saying that it switches between NSW and Boussinesq equations if "the area of simulation shows dispersive effects" is not sufficient. Please explain how this is done (on the basis of which criterion). For Telemac-2D and SCISM, eqs. are given but it is not clear how these are expressed in spherical coordinates (which is necessary for transoceanic propagation).

More details of the mu(I) rheology should be added (e.g., top of p4) as many readers may not be familiar with it. Also, the use of mu(I) appears more as an afterthought and it should be clearly said from the onset that 2 rheologies (Newtonian and mu(I)) were simulated for each source (say on p3 line 23).

Section 2.5 should be more about "Tsunami impact (or hazard) assessment". Hazard can be expressed as a function of runup, but also flow depth (maximum surface elevation), velocity, momentum force, etc... Regarding runup, the use of the analytical solution for a plane slope in areas of coarse grid seems far-fetched. As stated later in the paper, focusing-defocusing and friction effects, will dramatically change the runup estimated this way. I suggest eliminating this part entirely from the paper as it seems to this reviewer to bias the tsunami hazard assessment for CVV. Uniformly computing and comparing flow depth for some isobath (even at the shore) would be more meaningful.

Regarding friction, if this adjusted as a function of land use for some areas (e.g., Guadeloupe), this should be documented in the paper with a map of land use and a table of corresponding friction coefficient.

About slide results, l14 p10, as these are discussed in the text, slide cross- and along-sections should be provided in figures, plus runout contours for the various rheology cases so they can be compared.

[Figure]

Sentence on l24 p10 is fully implicit. Instead, one should explicitly discuss slide results and related wave generation, for each scenario, and then point out that wave elevations significantly vary, etc... Note the t=5 min (300s) is not a case in Fig. 11. Other figs are needed as discussed before. Saying l18 that MSL drastically lowers in the inviscid case as compared to the present viscous cases should be shown on a figure.

Fig. 12 (which by the way is highly distorted ?) shown in filter could be discussed in previous section when explicitly presenting results for slide and waves for each volume scenario.

L10 p11, again an example of an important discussion that should both be explicit and expanded. Say that Fig. 14 clearly shows well developed wavetrains illustrative of the high dispersive effects.

Some results cannot really be see on some figures which are poor in terms of color shade/scheme, brightness, etc.. Fig. 15 for instance does not allow seeing some "waves" (or amplitudes ? or surface elevations ?) are larger than 5 m as stated L20 p11. Fig. 17 is another one with poor color scheme. Also make sure that all the names of localities listed L25-31 p11 are marked on the figure in a visible manner (most are in Fig. 6), with a separate table providing their geographic coordinates. Table 1 caption is not explicit as of where gauges 1-6 are and what they are; it seems to be stations n Fig. 7, but then caption for each table/figure should refer to the other one.

L6-7 p 12 not only is a weak discussion regarding the SCHISM-FUNWAVE comparison in Fig 19 but is not quite right based on results. Here is a case where dispersive effects for long distance propagation of an initially identical train of (quite dispersive waves) are very well illustrated with dispersion causing a marked redistribution of the elevation/energy of the leading wave onto the 2nd and/or 3rd-4th waves. SCISM likely has some numerical dispersion that causes high frequency oscillations in the tail of the wavetrain compared to FUNWAVE. An energy spectrum would also be useful here in comparing where energy is located as a function of frequency. Differences between

model results in the tail also results from different wetting-drying algorithms schemes at the shore and dissipation in breaking waves that may cause vastly different reflected wave trains that interact with the incident wavetrain in the tail. FUNWAVE-TVD in particular features an improved shoreline algorithm following results of a benchmarking workshop. What is the status of the 3 other models in this respect ? None of this is properly discussed anywhere in the paper. In any case, this reviewer's opinion is that results in Fig. 19 show SCHISM is not applicable (or an appropriate) model for the transoceanic propagation of CVV wavetrains. Other results shown later confirm this opinion. A more adequate modeling, if SCHISM needs to be used nearshore, would have been to nest a SCHISM grid (around Guadeloupe or similar), within the ocean basin grid in which waves would have been simulated using a dispersive model.

Likewise, with Telemac-2D, which is non-dispersive, it is fully expected to see large discrepancies (as mentioned in L20 p12) of results with those of Calypso and FUNWAVE-TVD) in Figs. 21-24. Clearly the same conclusion as for SCISM applies to Telemac-2D, which should only be used in a nearshore nested domain to compute onshore inundation, but is not appropriate for the long distance propagation of dispersive wavetrains.

Statement in L26-28 p 12 is not apparent to this reviewer, unlike stated in the text. Be more explicit or use better support. L29- discussion of gauge 3; as indicated before, model results are also influenced by differences and implementation of breaking dissipation and wetting-drying shoreline algorithms (affecting reflection), and also open boundary conditions/sponge layers if any (not discussed here) that, if insufficient) may reflect some waves back into the domain.

L1-2 p13 is another case where a Fourier transform would help clarifying results and their interpretation.

In section 3.4, the runup discussion includes many coarse grid areas where comparing flow depth at the shore would have been sufficient. This reviewer's believes the analytical runup computation and discussion in 3.4.1 does not really apply or is useful here. I

suggest removing section 3.4.1. Another argument in support of the latter is statement on L9 p14 that the analytical formula overpredicts results due to lack of friction that "drastically influence(s) inundation patterns".

L12-13 p14, this work is tsunami hazard assessment not risk. The paper uses those terms alternately as if they were synonymous. They are not.

L24 p14. How can the extreme 450 km3 scenario be entirely ruled out by a simple statement and how is 80 km3 more believable. Day (personal communication) still thinks this is a potential collapse scenario should a large enough earthquake/eruption affect the island.

L5-25 p15, clearly for reasons of their own and certainly not based on results, the authors are trying to claim that all 4 models have equal skills in predicting the far-field impact of the CFF tsunami. This is clearly not the case and this discussion is quite moot. For this reviewer it is clear from results that SCHISM and Telemac 2D are not proper model to compute the propagation of the CVV tsunamis (eg see Figs 19, 21-24).

Differences between model results are not only due to resolution (L12 p15), but besides dispersive physics, also to the shoreline and breaking algorithms. I strongly disagree with the discussion of L20-23 p15 and conclusions of L13-15 p16 and L16-20 (see Figs. 19, 21-24). These conclusions are misleading and should not be a recommendation for the modeling community to adopt. Also, please compute the dispersion index of Glimsdal et al (2013) and see for yourself.

Editorial comments

Inconsistency of terminology in the text to refer to surface elevation, such as wave, wave amplitude, height, tsunami amplitude, etc... I suggest using surface (or maximum surface) elevation in the ocean up to the coast, then flow depth at the coast, and runup onshore. It should be also clearly defined what hazard and risk (i.e., exposure to hazard) mean and that these terms are not interchangeable. Use of a term indicating

return period should be uniform. The paper uses frequency or probability or probable or more frequent, source, etc.... This reviewer suggest long or short return period events.

There are many acronyms in the text, some of these not defined (e.g., SHOM). They should all be defined the first time they are used and I suggest adding a "table of acronyms" at the end. Note, accepted acronym for Nonlinear Shallow Water eqs. is usually NSW.

Figures have inconsistent format and color schemes and also some are difficult too read (i.e., very dark color scheme etc..). This is likely due to different modelers producing different figures with different software but it would really be important to improve the figures as they report the key findings of the work. Using a single software and color scheme to produce figures would be important.

There are some improper use in the text of English words and grammar that would warrant a final editing by a native speaker. Some rewordings are suggested below.

Fig. 3 caption, please replace eta by mu.

Figs. 4 and 5 (and also Fig. 6,7): use brighter more discriminant color scheme, also show a measure of bathymetry (contours). Label grids and give a table with grid characteristics.

Figs 8-9 are more qualitative than quantitative. Actual cross-sections (vertical and horizontal) should be shown with a metric scale.

Color scheme in Figs. 10-14 should also be used in following figures showing surface elevation rather than the red-pink/blue or blue/gray schemes. Also, in general, color scheme should be selected to make figures more readable in terms of having nicely spread out colors which does not occur in most figures which are based on the maximum value. Fig. 17 is a good example of this where one can't really see much of the tsunami elevation as the scheme goes to too high a maximum. Fig 17 is another

example of a difficult to read color scheme in the low values

Figs. 18,19: make figure more readable by removing the first 5h or so where nothing happens, and moving legend to caption.

This is the case in Fig. 15 where colors should be selected based on making the far-field impact more visible rather than having it in more or less a uniform color. Panel on the left is unreadable in this respect. Caption of Fig. 15 says 2.7 km resolution. Aren't these spherical coordinate grids for transoceanic propagation. This is not clear since grids are not clearly defined

Additional references

Glimsdal, S., Pedersen, G. K., Harbitz, C. B., & Løvholt, F. (2013). Dispersion of tsunamis: Does it really matter? Natural Hazards and Earth System Sciences, 13, 1507–1526. https://doi.org/10.5194/nhess-13-1507-2013.

Grilli, S.T., Grilli A.R., David, E. and C. Coulet 2016. Tsunami Hazard Assessment along the North Shore of Hispaniola from far- and near-field Atlantic sources. Natural Hazards, 82(2), 777-810, doi: 10.1007/s11069-016-2218-z Grilli S.T., D.R. Tappin, S. Carey, S.F.L. Watt, S.N. Ward, A.R. Grilli, S.L. Engwell, C. Zhang, J.T. Kirby, L. Schambach and M. Muin 2019. Modelling of the tsunami from the December 22, 2018 lateral collapse of Anak Krakatau volcano in the Sunda Straits, Indonesia, Scientific Reports, 9, 11946 (open access) doi:10.1038/s41598-019-48327-6.

Horrillo, J., Wood, A., Kim, G. B., & Parambath, A. 2013. A simplified 3‐D Navier‐Stokes numerical model for landslide‐tsunami: Application to the Gulf of Mexico. Journal of Geophysical Research: Oceans, 118(12), 6934-6950.

Kirby, J. T., Shi, F., Nicolsky, D. & Misra, S 2016. The 27 April 1975 Kitimat, British Columbia, submarine landslide tsunami: a comparison of modeling approaches. Landslides 13, 1421–1434, https://doi.org/10.1007/s10346-016-0682-x.

Løvholt, F., Bondevik, S., Laberg, J. S., Kim, J., & Boylan, N. 2017. Some giant subma-

rine landslides do not produce large tsunamis. Geophysical Research Letters, 44(16), 8463-8472.

Ma, G., Kirby, J. T., Hsu, T.-J. & Shi, F 2015. A two-layer granular landslide model for tsunami wave generation: Theory and computation. Ocean Modelling 93, 40–55, https://doi.org/10.1016/j.ocemod.2015.07.012.

Schambach L., Grilli S.T., Kirby J.T. and F. Shi 2019. Landslide tsunami hazard along the upper US East Coast: effects of slide rheology, bottom friction, and frequency dispersion. Pure and Applied Geophys., 176(7), 3,059-3,098,doi.org/10.1007/s00024-018-1978-7 (published online 09/03/18).

---

## Referee Comment (RC2) · Anonymous Referee #2 · 15 Dec 2019

Overall Assessment: The study is an important contribution to assessing the tsunami hazard to the coasts of countries exposed to the potential tsunami from the collapse of the Cumbre Vieja volcano at La Palma island. The study is not just useful for the scientific community but also to disaster managers of those countries so as to better prepare for any eventuality. I would like to recommend this manuscript provided the author addresses the following comments.

Comments: Pages 1-3: Section 1 on introduction should also mention modelling studies related to the Storegga slide event and anak krakatau December 2018 event. Moreover, an index map should also be provided showing the location of Cumbre Vieja volcano with reference to the potential areas that may be exposed to the ensuing tsunami in case of collapse. The index map should also show zoom in regions that are being analysed in the paper.

Page 5: What are the reasons for using THETIS over other available models in Section 2.2?

Pages 3-9: All the models discussed share the same basic equations, differences between those models must be presented in one single section. These differences may be with respect to assumptions, limitations or the numerical methods used. No need to show the equations.

Pages 3-9: Following section 2.3 on models used for long distance propagation, a separate section on the DEM should be provided . Also, figure showing elevations of the computational region must be provided. A sub-section on the grids utilised in each of the models can then be covered in this section.

Page 15 line 27: Limitation and assumptions of the current study should be discussed in this study.

Page 16: Conclusion should discuss steps that can be considered for improvement in better understanding the tsunami hazard.

Figures 8 and 9 are hard to follow especially interms of location and orientation of slide.

Figure 18 and 19: fonts are not clear.

---

## Author Comment (AC1) · 10 Feb 2020

Authors' responses:

The comments of the first reviewer concerned five main points listed below:

1. Slide and wave generation results: The requested snapshots showing quantitative (slide plan view and thickness evolution with time) and therefore reproducible results on the slide and the wave generation are now provided in Figure 8. (Note that run-out contours are not included because the slide is still flowing at the time the Navier-

[Figure]

Stokes simulation is stopped. Nevertheless, the figure provided fully covers the energetic transfer from slide to free surface). In addition to the existing SEANOE repository (DOI 10.17882/61301) which contains wave data for the source, we have also added a directory with the necessary information for interested readers to be able to perform their own simulations with the same initial conditions as the current paper.

2. Dispersive effects: We agree with the remarks of this reviewer on the role of dispersive effects. Therefore, the Boussinesq model FUNWAVE-TVD used for propagation is now presented as the reference model as it is the only one taking dispersive effects into account from the source to the coast. Only results obtained with this model are presented in the results section to provide quantitative information on the tsunami features on the selected gauges for this case. The other models are now presented in a specific section entitled "other models used" with the only objective to compare with the reference model and between each other.

3. Presentation of the propagation models: Equations have been removed, information on model resolution, coordinate type (spherical, Cartesian), Manning coefficient distribution, wetting-drying algorithms schemes at the shore, and breaking wave modeling is now provided. Purpose of the comparison is also clarified.

4. Wave impact and run-up: In La Guadeloupe, new SCHISM computations have been performed using FUNWAVE-TVD results off the island as an input signal (hotstart) as suggested by the reviewer. Maximum flow depth calculations at isobath -5 m are also presented for three areas along the French Atlantic coastline covered with fine grids. This calculation is based on the Boussinesq version of the Calypso code for propagation and shallow water version for inundation. We showed in the paper that the results obtained with this nested model agrees very well with FUNWAVE-TVD results.

Nevertheless, maximum run-up and flow velocity using the formula given by Madsen and Fuhrman (2008) are still provided but only as an additional tool to help the reader make a quick assessment of tsunami hazard where flow depth is not provided or true

inundation is not calculated.

5. Miscellaneous (typos, figure quality, references, English): The remarks from the reviewer have been taken into account.

Responses to specific comments and related text changes are indicated below.
* * *
Review comments: (specific responses from the authors in bold characters)

In introduction, the paper lists some earlier work on CVV tsunami modeling and far-field impact but does not really quantitatively summarize all the important findings. Some references also are missing, for instance Grilli et al. (2016).

Response : This part has been modified accordingly, and references added.

In introduction, the paper should better discuss why various propagation models are used to simulate the same cases (for instance to assess epistemic uncertainty). It seems that models were used only based on their availability or the experience within a certain group and not because they really featured the necessary physics.

Response : Done - see first paragraph section "Other models used".

As early as Løvholt et al. (2008), it was made clear that CVV collapses generate highly dispersive wavetrains. This was also reported in the far-field by Tehranirad et al. (2015). Here, both dispersive and non-dispersive models are used and although some of the results presented (particularly in and around Guadeloupe) clearly show significant dispersive effects that in a standard way reduce the leading wave elevation and enhance that of the second and/or third waves and then shed a larger tail of oscillation, the conclusion is more or less that all models are acceptable to simulate this event, as this were a forgone conclusion ?

Response : See new results and conclusions on dispersive effects in section 3.3, 3.4, 4 and 5.

Also, the conclusions wrongly imply that dispersive effects only express themselves early in the event, during the THETIS and FUNWAVE near-field simulations which are both dispersive, and hence it is fine using a non-dispersive model for the transoceanic propagation. This contrary to established effects of dispersion, which a train of short enough waves, will go growing with time and distance. See for instance the discussion and dispersion indicators (which could be computed here) detailed in Glimsdal et al. (2013), and also applied in Schambach et al (2019) for landslide tsunamis. In this reviewer's opinion, the results presented here would lead to conclude using TELEMAC-2D or SCHISM for anything other then the nearshore inundation should be ruled out. See for instance Grilli et al. (2016) for which the transoceanic propagation of the CVV tsunami was done with FUNWAVE and the local impact on Hispaniola was computed with TELEMAC-2D in an unstructured grid.

Response : In the new manuscript, the effect of dispersive effect is acknowledged and illustrated for instance in Figure 21. As well, this sentence has been added to the conclusion: "After 15 minutes of propagation in a Boussinesq model, the wave signal is still dispersive and therefore Boussinesq models should be recommended to use the source provided."

Here the authors present new, more realistic, CVV sources, but the comparison between the old sources and the new ones is quite limited. More details would be desirable for both the wave generation and the slide. To begin with, the initial topography and geometry used for each source (both old and new ones and all the considered volumes), such as shown in Fig. 1, should be better described (is it fully available on some data storage site ?).

Response : In addition to the existing SEANOE repository (DOI 10.17882/61301) which contains wave data for the source, we have added a directory with the necessary information for interested readers to be able to perform their own simulations with the same initial conditions as the current paper.

Then, it would be desirable to clearly show stages of the slide development underwater, for the old inviscid source and the new ones. In this respect, Figs. 8 and 9 are hard to get any information from. Having both comparative cross section and planview of slide flow and runout as a function of time would be desirable. For the generated tsunami, only Fig. 10 shows some measure of comparison but there as well more comparison allowing the reader to understand the nature of the tsunami source and how it is affected by viscosity would be helpful. P5 l8, it is mentioned the landslide continues to move for a very long time as in Abadie et al (2012) but this should be illustrated in a figure.

Response : See Figure 8 and associated text.

There is no real review of other models that have been used for simulating subaerial or submarine landslide tsunamis, for dense fluid or granular mediums. I suggest for instance to refer to and at least briefly discuss models of Horrillo et al. (2013), Ma et al. (2015) and Kirby et al. (2016) and Løvholt et al. (2017). See also the recent application of the Ma et al. and Kirby et al. models to a volcanic collapse tsunami in Grilli et al. (2019).

Response : Review added in the third paragraph of section 2.1.

Model grids are not clearly enough defined. Better (more readable, less dark) figures of grids on a global map, plus some regional zoom outs, and tables of parameters are required.

Response : Better and lighter figures with label grids, and a table with grids parameters were added.

Lines 22-29 p 5 are too implicit. For instance, for SCHISM it is said that "the resolution is adapted to accurately reproduce wavetrains of 2 minutes or more with at least 20 nodes per wavelength in deep ocean". Is this applicable to all 4 models ? How was this decided about ? Please explicitly explain models and grid used and how.

[Figure]

Response : Done – see section "other models used".

L32 p5, saying all models use the same basic equations is meaningless. Please specify common physical assumptions and differences among models. Listing eqs. for 3 out of 4 of the models without background is meaningless. Equations are not necessary here but model physics, features and limitations are. If the authors do want to list eqs., they should do it for all models and in appendix.

Response : As already mentioned, the presentation of the model has been modified in this new version. Equations have been removed and main differences between models clarified.

The end of section 2.3 would be a good discuss anticipated dispersive effects based on earlier work on landslide tsunamis and on CVV. Regarding Calypso, saying that it switches between NSW and Boussinesq equations if "the area of simulation shows dispersive effects" is not sufficient. Please explain how this is done (on the basis of which criterion).

Response : Done – see section 2.2.2 "other models used".

For Telemac-2D and SCISM, eqs. Are given but it is not clear how these are expressed in spherical coordinates (which is necessary for transoceanic propagation).

Response : This aspect is clarified - see section "Other models used".

More details of the mu(I) rheology should be added (e.g., top of p4) as many readers may not be familiar with it. Also, the use of mu(I) appears more as an afterthought and it should be clearly said from the onset that 2 rheologies (Newtonian and mu(I)) were simulated for each source (say on p3 line 23).

Response : Done – see section 2.1 "Navier-Stokes simulation of wave source".

Section 2.5 should be more about "Tsunami impact (or hazard) assessment".

Response : Risk replaced by hazard as suggested.

Hazard can be expressed as a function of runup, but also flow depth (maximum surface elevation), velocity, momentum force, etc. . . Regarding runup, the use of the analytical solution for a plane slope in areas of coarse grid seems far-fetched. As stated later in the paper, focusing-defocusing and friction effects, will dramatically change the runup estimated this way. I suggest eliminating this part entirely from the paper as it seems to this reviewer to bias the tsunami hazard assessment for CVV. Uniformly computing and comparing flow depth for some isobath (even at the shore) would be more meaningful.

Response : Flow depths at isobath -5m has been computed at three locations along the French Atlantic coastline. See Figure 25.

Regarding friction, if this adjusted as a function of land use for some areas (e.g., Guadeloupe), this should be documented in the paper with a map of land use and a table of corresponding friction coefficient.

Response : New figure provided (a table would be too large to be provided).

About slide results, l14 p10, as these are discussed in the text, slide cross- and along - sections should be provided in figures, plus runout contours for the various rheology cases so they can be compared.

Response : See the new Figure 8.

Sentence on l24 p10 is fully implicit. Instead, one should explicitly discuss slide results and related wave generation, for each scenario, and then point out that wave elevations significantly vary, etc. . .

Note the t=5 min (300s) is not a case in Fig. 11.

Other figs are needed as discussed before.

Response : All the information requested by the reviewer has been summarized in the new Figure 8.

Saying l18 that MSL drastically lowers in the inviscid case as compared to the present

viscous cases should be shown on a figure.

Response : This is now fully observable on the new Figure 8.

Fig. 12 (which by the way is highly distorted ? - Response : wrong figure aspect ratio fixed.

shown in filter could be discussed in previous section when explicitly presenting results for slide and waves for each volume scenario.

Response : The authors decided not to explicitly present and discuss all the different slide scenarios but rather discuss them implicitly as the paper has already many figures, and snapshots would show very similar pattern but of lower amplitudes (which is indicated in the text).

L10 p11, again an example of an important discussion that should both be explicit and expanded. Say that Fig. 14 clearly shows well developed wavetrains is illustrative of the high dispersive effects.

Response : Dispersive effect is fully acknowledged and discussed in this new version.

Some results cannot really be seen on some figures which are poor in terms of color shade/scheme, brightness, etc.. Fig. 15 for instance does not allow seeing some "waves" (or amplitudes ? or surface elevations ?) are larger than 5 m as stated L20 p11.

Response : Done.

Fig. 17 is another one with poor color scheme. Also make sure that all the names of localities listed L25-31 p11 are marked on the figure in a visible manner (most are in Fig. 6), with a separate table providing their geographic coordinates. Table 1 caption is not explicit as of where gauges 1-6 are and what they are; it seems to be stations n Fig. 7, but then caption for each table/figure should refer to the other one.

Response : New figure has been provided.

L6-7 p 12 not only is a weak discussion regarding the SCHISM-FUNWAVE comparison in Fig 19 but is not quite right based on results. Here is a case where dispersive effects for long distance propagation of an initially identical train of (quite dispersive waves) are very well illustrated with dispersion causing a marked redistribution of the elevation/energy of the leading wave onto the 2nd and/or 3rd-4th waves. SCISM likely has some numerical dispersion that causes high frequency oscillations in the tail of the wavetrain compared to FUNWAVE. An energy spectrum would also be useful here in comparing where energy is located as a function of frequency.

Response : Fourier transforms are now provided for almost every gauge and used for discussion of dispersive effects.

Differences between model results in the tail also results from different wetting-drying algorithms schemes at the shore and dissipation in breaking waves that may cause vastly different reflected wave trains that interact with the incident wavetrain in the tail. FUNWAVE-TVD in particular features an improved shoreline algorithm following results of a benchmarking workshop. What is the status of the 3 other models in this respect ? None of this is properly discussed anywhere in the paper.

Response : See new section other model used regarding these questions.

In any case, this reviewer's opinion is that results in Fig. 19 show SCHISM is not applicable (or an appropriate) model for the transoceanic propagation of CVV wavetrains. Other results shown later confirm this opinion. A more adequate modeling, if SCHISM needs to be used nearshore, would have been to nest a SCHISM grid (around Guadeloupe or similar), within the ocean basin grid in which waves would have been simulated using a dispersive model.

Response : Done: tsunami hazard assessment in Guadeloupe is now performed by SCHISM using FUNWAVE-TVD as a hotstart.

Likewise, with Telemac-2D, which is non-dispersive, it is fully expected to see large discrepancies (as mentioned in L20 p12) of results with those of Calypso and FUNWAVE-TVD) in Figs. 21-24. Clearly the same conclusion as for SCISM applies to Telemac-2D, which should only be used in a nearshore nested domain to compute onshore inundation, but is not appropriate for the long distance propagation of dispersive wavetrains.

Response : The use of NSW models for propagation is now clearly justified as a benchmarking exercise to quantify the respective effect of dispersion and resolution, as further specified in the conclusions.

Statement in L26-28 p 12 is not apparent to this reviewer, unlike stated in the text. Be more explicit or use better support. L29- discussion of gauge 3; as indicated before, model results are also influenced by differences and implementation of breaking dissipation and wetting-drying shoreline algorithms (affecting reflection), and also open boundary conditions/sponge layers if any (not discussed here) that, if insufficient) may reflect some waves back into the domain.

L1-2 p13 is another case where a Fourier transform would help clarifying results and their interpretation.

Response : Done.

In section 3.4, the runup discussion includes many coarse grid areas where comparing flow depth at the shore would have been sufficient. This reviewer's believes the analytical runup computation and discussion in 3.4.1 does not really apply or is useful here. I suggest removing section 3.4.1. Another argument in support of the latter is statement on L9 p14 that the analytical formula overpredicts results due to lack of friction that "drastically influence(s) inundation patterns".

Response : As already indicated, maximum run-up and flow velocity formula given by Madsen and Fuhrman (2008) are still provided, but only as an additional tool to help the reader make a quick assessment of tsunami hazard where flow depth is not provided, or where actual inundation is not calculated.

L12-13 p14, this work is tsunami hazard assessment not risk. The paper uses those terms alternately as if they were synonymous. They are not.

Response : Done.

L24 p14. How can the extreme 450 km3 scenario be entirely ruled out by a simple statement and how is 80 km3 more believable. Day (personal communication) still thinks this is a potential collapse scenario should a large enough earthquake/eruption affect the island.

Response : We agree that this scenario cannot be ruled out. See the new section of remarks in the discussion.

L5-25 p15, clearly for reasons of their own and certainly not based on results, the authors are trying to claim that all 4 models have equal skills in predicting the far-field impact of the CFF tsunami. This is clearly not the case and this discussion is quite moot. For this reviewer it is clear from results that SCHISM and Telemac 2D are not proper model to compute the propagation of the CVV tsunamis (eg see Figs 19, 21-24).

response : See corresponding new text in the discussion.

Differences between model results are not only due to resolution (L12 p15), but besides dispersive physics, also to the shoreline and breaking algorithms. I strongly disagree with the discussion of L20-23 p15 and conclusions of L13-15 p16 and L16-20 (see Figs. 19, 21-24). These conclusions are misleading and should not be a recommendation for the modeling community to adopt.

Response : Recommendations and conclusions changed in the new version.

Also, please compute the dispersion index of Glimsdal et al (2013) and see for yourself.

Response : See discussion on wave frequencies present in the spectrum after 15 min. of propagation in Section 3.3: "Propagation : FUNWAVE-TVD Results".

Editorial comments Inconsistency of terminology in the text to refer to surface elevation,

such as wave, wave amplitude, height, tsunami amplitude, etc. . . I suggest using surface (or maximum surface) elevation in the ocean up to the coast, then flow depth at the coast, and runup onshore. It should be also clearly defined what hazard and risk (i.e., exposure to hazard) mean and that these terms are not interchangeable. Use of a term indicating return period should be uniform. The paper uses frequency or probability or probable or more frequent, source, etc. . .. This reviewer suggest long or short return period events.

response : Done whenever possible.

There are many acronyms in the text, some of these not defined (e.g., SHOM). They should all be defined the first time they are used and I suggest adding a "table of acronyms" at the end. Note, accepted acronym for Nonlinear Shallow Water eqs. Is usually NSW.

Response : All the NLSW acronyms have been replaced by NSW. All the acronyms are now clearly defined.

Figures have inconsistent format and color schemes and also some are difficult too read (i.e., very dark color scheme etc..). This is likely due to different modelers produc-ing different figures with different software but it would really be important to improve the figures as they report the key findings of the work. Using a single software and color scheme to produce figures would be important.

Response : Done whenever possible.

There are some improper use in the text of English words and grammar that would warrant a final editing by a native speaker.

Response : Revised manuscript checked by a native speaker.

Some rewordings are suggested below.

Fig. 3 caption, please replace eta by mu.

response : Done.

Figs. 4 and 5 (and also Fig. 6,7): use brighter more discriminant color scheme, also show a measure of bathymetry (contours). Label grids and give a table with grid characteristics.

Response : The maps have been redrawn (Figures 4, 5 and 6-new) with a different software, and a table with grid characteristics has been added. Figure 6-old has been deleted, and the gauges that it indicated have been added to Figures 4 and 5.

Figs 8-9 are more qualitative than quantitative. Actual cross-sections (vertical and horizontal) should be shown with a metric scale. Color scheme in Figs. 10-14 should also be used in following figures showing surface elevation rather than the red-pink/blue or blue/gray schemes.

Response : As suggested, color schemes in Figures 15-new and 19-new have been adapted from color scheme in Figures 10-14-old.

Also, in general, color scheme should be selected to make figures more readable in terms of having nicely spread out colors which does not occur in most figures which are based on the maximum value.

Fig. 17 is a good example of this where one can't really see much of the tsunami elevation as the scheme goes to too high a maximum. Fig 17 is another example of a difficult to read color scheme in the low values

Response : Done.

Figs. 18,19: make figure more readable by removing the first 5h or so where nothing happens, and moving legend to caption.

Response : Done.

This is the case in Fig. 15 where colors should be selected based on making the far-field impact more visible rather than having it in more or less a uniform color. Panel on

the left is unreadable in this respect.

Response : Done.

Caption of Fig. 15 says 2.7 km resolution. Aren't these spherical coordinate grids for transoceanic propagation. This is not clear since grids are not clearly defined

Response : Information on coordinate system added in the model section.

Additional references Glimsdal, S., Pedersen, G. K., Harbitz, C. B., & Løvholt, F. (2013). Dispersion of tsunamis: Does it really matter? Natural Hazards and Earth System Sciences, 13, 1507–1526. https://doi.org/10.5194/nhess-13-1507-2013. Grilli, S.T., Grilli A.R., David, E. and C. Coulet 2016. Tsunami Hazard Assessment along the North Shore of Hispaniola from far- and near-field Atlantic sources. Natural Hazards, 82(2), 777-810, doi: 10.1007/s11069-016-2218-z Grilli S.T., D.R. Tappin, S. Carey, S.F.L. Watt, S.N. Ward, A.R. Grilli, S.L. Engwell, C. Zhang, J.T. Kirby, L. Schambach and M. Muin 2019. Modelling of the tsunami from the December 22, 2018 lateral collapse of Anak Krakatau volcano in the Sunda Straits, Indonesia, Scientific Reports, 9, 11946 (open access) doi:10.1038/s41598-019-48327-6. Horrillo, J., Wood, A., Kim, G. B., & Parambath, A. 2013. A simplified Navier-Stokes numerical model for landslide-tsunami: Application to the Gulf of Mexico. Journal of Geophysical Research: Oceans, 118(12), 6934-6950. Kirby, J. T., Shi, F., Nicolsky, D. & Misra, S 2016. The 27 April 1975 Kitimat, British Columbia, submarine landslide tsunami: a comparison of modeling approaches. Land-slides 13, 1421–1434, https://doi.org/10.1007/s10346-016-0682-x. Løvholt, F., Bondevik, S., Laberg, J. S., Kim, J., & Boylan, N. 2017. Some giant submarine landslides do not produce large tsunamis. Geophysical Research Letters, 44(16), 8463-8472. Ma, G., Kirby, J. T., Hsu, T.-J. & Shi, F 2015. A two-layer granular landslide model for tsunami wave generation: Theory and computation. Ocean Modelling 93, 40–55, https://doi.org/10.1016/j.ocemod.2015.07.012. Schambach L., Grilli S.T., Kirby J.T. and F. Shi 2019. Landslide tsunami hazard along the upper US East Coast: effects of slide rheology, bottom friction, and frequency dispersion. Pure and Applied

[Figure]

Geophys., 176(7), 3,059-3,098,doi.org/10.1007/s00024-018-1978-7 (published online 09/03/18).

All the above references except the first and the last one are now quoted in the manuscript.

Please also note the supplement to this comment:
https://www.nat-hazards-earth-syst-sci-discuss.net/nhess-2019-225/nhess-2019-225-AC1-supplement.pdf

---

## Author Comment (AC2) · 10 Feb 2020

Overall Assessment: The study is an important contribution to assessing the tsunami hazard to the coasts of countries exposed to the potential tsunami from the collapse of the Cumbre Vieja volcano at La Palma island. The study is not just useful for the scientific community but also to disaster managers of those countries so as to better

prepare for any eventuality. I would like to recommend this manuscript provided the author addresses the following comments.

Comments: Pages 1-3: Section 1 on introduction should also mention modelling studies related to the Storegga slide event and anak krakatau December 2018 event.

Response : The following references on the Anak Krakatau are now quoted in the introduction:

Paris, A., Heinrich, P., Paris, R., and Abadie, S.: The December 22, 2018 Anak Krakatau, Indonesia, Landslide and Tsunami: Preliminary Modeling Results, Pure and Applied Geophysics, pp. 1–20, 2019.

Grilli, S. T., Tappin, D. R., Carey, S., Watt, S. F., Ward, S. N., Grilli, A. R., Engwell, S. L., Zhang, C., Kirby, J. T., Schambach, L., et al.: Modelling of the tsunami from the December 22, 2018 lateral collapse of Anak Krakatau volcano in the Sunda Straits, Indonesia, Scientific reports, 9, 2019.

Regarding the Storrega slide, two additional references are quoted in the new paragraph on landslide tsunami models provided in section 2.1, namely:

Løvholt, F., Bondevik, S., Laberg, J. S., Kim, J., and Boylan, N.: Some giant submarine landslides do not produce large tsunamis, Geophysical Research Letters, 44, 8463–8472, 2017

Kim, J., Løvholt, F., Issler, D., and Forsberg, C. F.: Landslide Material Control on Tsunami Genesis—The Storegga Slide and Tsunami (8,100 Years BP), Journal of Geophysical Research: Oceans, 2019.

Moreover, an index map should also be provided showing the location of Cumbre Vieja volcano with reference to the potential areas that may be exposed to the ensuing tsunami in case of collapse. The index map should also show zoom in regions that are being analysed in the paper.

Response : Done: See Figures 4 to 7.

Page 5: What are the reasons for using THETIS over other available models in Section 2.2?

Response : A review of the most advanced models for landslide tsunami generation is now provided in section 2.1 for qualitative inter-comparison with THETIS. As a full 3D Navier-Stokes model with 3 phases, THETIS is clearly one of the most advanced types of models, although the CPU time required is substantial. The authors' team has more than 20 years of experience with this kind of model, though more frequently used for small scale fluid mechanics problems than in geophysical flows of large scale. Moreover, as stated now in the text (see section 2.1), "THETIS has been validated against several benchmark cases involving tsunami generated by 2D and 3D solid blocks (Abadie et al., 2010) and granular subaerial and submarine slides (Clous and Abadie, 2019)Âă".

Pages 3-9: All the models discussed share the same basic equations, differences between those models must be presented in one single section. These differences may be with respect to assumptions, limitations or the numerical methods used. No need to show the equations.

Response : Equations have been removed, information on model resolution, coordinate type (spherical, Cartesian), Manning coefficient distribution, wetting-drying algorithms schemes at the shore, and breaking wave modeling is now provided. The purpose of comparison is also clarified.

Pages 3-9: Following section 2.3 on models used for long distance propagation, a separate section on the DEM should be provided. Also, figure showing elevations of the computational region must be provided. A subsection on the grids used in each of the models can then be covered in this section.

Page 15 line 27: Limitation and assumptions of the current study should be discussed

in this study.

Page 16: Conclusion should discuss steps that can be considered for improvement in better understanding the tsunami hazard.

Response : Limitations and improvements are now discussed more in details at the end of the discussion section.

Figures 8 and 9 are hard to follow especially in terms of location and orientation of slide.

Response : Snapshots showing quantitative (slide plan view and thickness evolution with time) and therefore reproducible results on the slide and the wave generation are now provided in Figure 8. In addition to the existing SEANOE repository (DOI 10.17882/61301) which contains wave data for the source, we have added a directory with the necessary information for interested readers to be able to perform their own simulations with the same initial conditions as the current paper.

Figure 18 and 19: fonts are not clear.

Response : Those figures have been redrawn with better fonts (Figures 17-new and 18-new).

Please also note the supplement to this comment:
https://www.nat-hazards-earth-syst-sci-discuss.net/nhess-2019-225/nhess-2019-225-AC2-supplement.pdf

---

## Author Response (AR4)

This paper is an elaborate study of the effect of three scenarios due to slope failure from La Palma Island on tsunami impact in French territories. It reports large amount of analysis and work, and there has been surely large efforts behind producing these outputs, in particular given that some of the tools employed such as the CFD THETHIS code are demanding to operate for such purposes. It is an important study because of the practical implications. On the other hand, the elaboration is also a drawback of the paper. There are many different models used, to illuminate different types of physics, merged with an attempt to make an impact assessment (the authors uses the term hazard). Moreover, the paper seems to have undergone several previous reviews with large changes, and would benefit from a better organisation.

Authors response: Indeed, we were very surprised not to see our revised paper ending up with the initial reviewers. It is obviously more difficult to please a reviewer which did not ask for the specific changes made on the initial submission.

Some related general comments are summarised briefly below, followed by a long list of line-byline comments. These comments must be taken into account in a possible revision of the manuscript.

**General comments:**

It is not clear why mane different models are used for various purposes. I would have liked a simpler strategy where the authors choose a simple set of models. The physics is well known: the tsunamis are dispersive, and we need nonlinear shallow water models for the inundation. Right now, there is a patchwork of models, even including analytical solutions (which I suggest to remove), and it is hard to understand why a given model is used where. While I would suggest that this is much simplified, I would probably expect that the authors would like to keep as much as possible of these results. Hence, as a minimum, a much tightened up introduction is needed to much better explain the scope and how the different models are used, and why. I would also suggest to better distinguish impact studies and studies of physical effects (e.g. dispersion, model comparisons).

**Authors response:**

A much simpler strategy has been finally adopted following this remark. Non dispersive long distance tsunami computations (i.e., with modells TELEMAC2D and SCHISM) have been removed from the manuscript. SCHISM is now only used for the impact assessment in La Guadeloupe taking the FUNWAVE-TVD signal as hotstart.

We kept the comparison between two Boussinesq models (FUNWAVE-TVD and calypso) off the french coastline to respect the spirit of the TANDEM project which was at the origin of this paper. Nevertheless, this corresponding portion of the paper is very short.

The impact study section has also been separated from the signal analysis part as requested by the reviewer.

We thank the reviewer to open the door for a doable revision for this matter. Indeed, this work was performed in the framework of a national project gathering several French

institutes developing or using different models. One of the underlying principle of this paper was to compare models and advertise the work performed within this project and therefore not to exclude anyone (also for political reasons). This principle makes the organization of the paper a bit difficult as maybe its reading. Considering the comment of this reviewer, we explained in the new version of the introduction this project aspect and defend its interest. [This response is labeled Response (1) for the next similar questions]

See paragraphs added at the end of the introduction (p3, I3):

"Computations performed by Gisler et al. (2006) or Abadie et al. (2012) were both based on inviscid or quasi-inviscid slide flow. In the present paper, the computations carried out in Abadie et al. (2012) are redone, improving their accuracy by calibrating the slide fluid viscosity in order to approach a granular slide (Sections 2.1 and 3.1) with a Newtonian model. Then, the same filtering process as in Abadie et al. (2012) is applied with the new wave sources to produce a wave signal which can be propagated by depth-averaged models (Sections 2.4 and 3.2). The three wave sources are then propagated using FUNWAVE-TVD (Section 2.2.1) and the results in the Caribbean Sea, in Western Europe and in France (Section 3.3) analyzed.

One of the goal of the TANDEM program was also the comparison of the models developed or used by the different partners of the project namely: Calypso developed by CEA, Telemac2D developed by EDF, Funwave-TVD used by BRGM\_and SCHISM by Université des Antilles. Here we take the opportunity of this case study to compare models on a real case and analyze the differences. The interest is double. This project involves partners who are already in charge of tsunami hazard assessment while others may play a role in this field at the national level in the future. The first interest is to provide an inter-comparison of the codes used at in the different institutes. This comparison will be valuable for future operational use. On the other hand, this comparison is made on a real case, therefore including all the inherent complexity and uncertainties (bathymetry, mesh, numerical parameters, physical parameters, etc.) usually associated to a practical case. Such a comparison is rarely attempted in usual benchmark exercises which focus more frequently on specific processes such as run-up, tsunami generation, etc..in order to make the interpretation easier. Nevertheless, even though the analysis is not straightforward because models are not based on the same assumptions, numerical methods, mesh types, a comparison including all the complexity may also be of interest as it allows to judge all the effect at once and potentially lead to practical recommendations valuable for future studies. Therefore the originality of this comparison on a real case is the second interest of this part of the study. Accordingly, the rest of the study is organized around a comparison of the different model results (see Sections 2.2.2 for description and 3.4 for the results comparison). Finally, tsunami impact is assessed in different areas in Section 3.5, and results interpreted and discussed in Section 4".

Another major issue, in particular when reading the introduction, is that you sense that the hazard study is attempting to make a best estimate of a landslide motion and wave generation based on laboratory glass bead experiments. However, nature will not behave this way, and there is a considerable uncertainty related to the process and the sliding material. Granted, one cannot perhaps expect that the computations can cover all these uncertainties, but as a minimum, the authors must make it crystal clear that there can be a much larger variability related to the tsunami generation and tsunamigenic strength. This is a limitation of the study.

**Authors response: We acknowledge this limitation. [This response is labeled Response (2) for the next similar questions]**

**See new paragraph added in the discussion on that matter:**

Second, we used a glass beads based experiment (Viroulet et al., 2013) to calibrate the Navier-Stokes simulation of the La Palma slide. If this is an improvement compared to the very coarse inviscid initial estimation (Abadie et al., 2012), which should be more considered as a worst case, such a laboratory experiment still is a huge simplification of the complexity expected in areal volcano collapse. An accurate description of such a complex process at real scale is still beyond the capabilities of current models. Therefore, there is here a very important source of uncertainty which the reader has to be aware of and this uncertainty propagates and affects the impact results. Furthermore, this work is not an hazard study which could have been performed for instance by considering different values of slide viscosity but at much higher computational cost. The position of this paper is rather to give an illustration of what could be expected from such an event by presenting results at least consistent with the current state of the art in terms of laboratory experiments and therefore propose an improvement compared to the previous published results on that case.

Finally, the title tsunami hazard is misleading, because the authors do not address return periods, in addition to lacking a proper treatment of the variability or sensitivity to landslide parameters as noted above. The title should hence be revised to take this into account.

*Authors response:* We agree with the reviewer that the term hazard was used inappropriately in the initial version of the paper. It has been removed when possible from the manuscript (except at the beginning of the introduction). Moreover we have added a paragraph in the discussion section about this limitation and one of the reference suggested below (Grezio et al., 2017) [This response is labeled Response (3) for the next similar questions]

See new paragraph added in the discussion on that matter:

Of course there are some limitations in this study which may provide the basis for future improvements. First, this study should not be considered as a hazard assessment stricto-sensu because the return period aspect is not considered and the sensitivity in the landslide parameters not covered extensively. For a review on Probabilistic Tsunami Hazard Analysis (PTHA) methods, the reader is referred to Grezio et al. (2017) for instance. Instead, the current study presents plausible particular scenarios based on state-of-the-art numeral models. Note that the Navier-Stokes model, which provides interesting information for this kind of processes, is still too heavy to be employed in PTHA computations.

**Detailed comments:**

Title: Probably the term "tsunamigenic strength from potential events" is better than hazard. After all, hazard refer to a temporal component, and should not really be used if return periods are not considered.

**Authors response: See response (3)**

New title: La Palma landslide tsunami: computation of the tsunami source with a calibrated multi-fluid-Navier-Stokes model, impact assessment, and model intercomparison New title : La Palma landslide tsunami: calibrated wave source and assessment of impact in French territories.

Page 1 line 5: "for 5 minutes" --> "after 5 minutes". *Authors response:* **Done**

Page 2 line 8: "allow studying impact on France and Guadeloupe". Here you maybe emphasise more strongly that this is the scope? After all, the impact locally would be a more natural focus. Page 1 - line 8: "Although the wave source seems to be reduced due to the rheology..." --> "Although the rheology applied in this study seemingly leads to smaller waves..." *Authors response:* Correction made (see point right after):

**Although the slide modeling approach applied in this study seemingly leads to smaller waves**

**Page 1 - line 9: add "mu(I)" ahead of rheology – *Authors response:* Not appropriate. The approach used here is a calibration of a Newtonian model – the mu(I) is just used once in this paper to justify this approach, hence the correction made (point right before).**

Page 2 - line 7: It the term hazard is used properly, it would be useful to introduce a definition, and refer to at least one key paper. Use e.g. Grezio et al. (2017): Grezio, A., Babeyko, A., Baptista, M. A., Behrens, J., Costa, A., Davies, G., ... & Harbitz, C. B. (2017). Probabilistic tsunami hazard analysis: Multiple sources and global applications. Reviews of Geophysics, 55(4), 1158-1198. Page 2 - line 13: On the complexity of these processes, please refer key review papers, Løvholt et al. (2015), Yavari-Ramshe and Ataie-Ashtiani (2016): Løvholt, F., Pedersen, G., Harbitz, C. B., Glimsdal, S., & Kim, J. (2015). On the characteristics of landslide tsunamis. Philosophical

Transactions of the Royal Society A: Mathematical, Physical and Engineering Sciences, 373(2053), 20140376; Yavari-Ramshe, S., & Ataie-Ashtiani, B. (2016). Numerical modeling of subaerial and submarine landslide-generated tsunami waves—recent advances and future challenges. Landslides, 13(6), 1325-1368.

**Authors response: See response (3)**

Page 3 line 9: Please update this sentence to say that you use mu(I). I think this is clearer than saying calibrated slide viscosity. *Authors response:* As stated before, the mu(I) rheology is only used in one simulation in this paper. So mu(I) is not added in the sentence but in place, the sentence has been changed to be clearer on that point.

Initial: In the present paper, the computations carried out in Abadie et al. (2012) are redone, improving their accuracy by calibrating the slide fluid viscosity in order to better represent a granular slide (Sections 2.1 and 3.1)

changed to: In the present paper, the computations carried out in Abadie et al. (2012) are redone, improving their accuracy by calibrating the slide fluid viscosity in order to approach a granular slide (Sections 2.1 and 3.1) with a Newtonian model.

Page 3 - line 29: "first instance of the motion" --> "initial motion". *Authors response:* Done

Page 3 - line 30: "solver code category" --> "type of solver". *Authors response:* Done

Page 3 - line 33: "close but not completely equivalent to models, also use to simulate landslide tsunami generation" --> "more sophisticated with respect to the slide motion than models such as" *Authors response:* Done

Page 4 - line 9: For a complete review, discuss also the model of Si et al 2018: This model is more sophisticated material wise, but probably not able to tackle operational environments yet: Si, P., Shi, H., & Yu, X. (2018). Development of a mathematical model for submarine granular flows. Physics of Fluids, 30(8), 083302. *Authors response:* Reference added and discussed as requested.

**Added:**

Finally, Eulerian-Eulerian two-phase models such as the one described in Si et al. (2018b) and Si et al. (2018a) are very promising approaches able to describe the flow within the grains as well as the grain/grain interactions but their applicability to practical cases has not been demonstrated yet.

Page 4 line 22: Clarify where Newtonian and mu(I) rheologies are used, maybe reformulate: "Both Newtonian and mu(I) rheologies are used in the simulations". *Authors response:* Done

Page 4 - line 23: The experimental results cannot necessarily represent the real case realistically (glass beads are far from a realistic rock slope material). Hence, all the different viscosities may represent the reality, and should not be calibrated towards a single dataset. This is actually a misconception, the hazard analysis should ideally include this as an uncertainty. Hence please reformulate. *Authors response:* done and see response (2)

Page 5 - first paragraph. Please see above comment. I dont believe a single calibrated result represent the reality realistically. This does not mean that new simulations should be done, but the authors should make the reader aware of this uncertainty. *Authors response:* see response (2)

Page 5 - line 15: Please explain that this is just possible value for the material parameter, and there

is likely a rather large uncertainty that is not covered in our analysis. Otherwise, the reader gets the false impression that the wave generation is deterministic, which it is'nt. *Authors response:* **Paragraph modified accordingly and see also response (2)**

To extrapolate these results for the La Palma computations, the following reasoning is adopted. First, it is assumed that the real slide is well represented by the granular medium used in the experiment. This approach is not deterministic as there is important differences between this experiment and the real case but at least it may be considered as a better assumption than the worst case scenario presented in Abadie et al. (2012)

Page 7 - line 3: Please clarify "can be upgraded"? Do you mean that it also contain dispersive features. In this case reformulate. It is BTW not clear why two types of dispersive models are used. Does this code have wetting and drying facilities?

**Authors response: Sentence reformulated:**

The user can choose to solve either the non-dispersive (NSW) or dispersive (Boussinesq model following Pedersen and Løvholt (2008)) non-linear long wave equations, written in spherical coordinates.

As explained in the text, the switch between non-dispersive and dispersive equations is realized between mother and daughter grids.

Yes, the code has wetting and drying facilities. It has been added in the text:

The wave impact assessment is realized using this mixed method for the French coasts and calculating run-up with wet and dry conditions. (removed from the current version)

Page 7 - line 25: Again, why is this model used? It is not clear why so many seemingly similar models are used, please elaborate.

Authors response: see response (1)model removed

Page 7 - line 34: "In this work..." do you refer to Telemac? The meaning is not clear. *Authors response:* model removed replaced by:

In this work, the mesh used in Telemac-2D has 12.5 million of...

Page 9 - line 17: This is not a proper hazard assessment. Impact analysis or scenario analysis are better terms.

**Authors response: see response (3)**

Page 9 - line 22: Again, I miss the reasoning for choosing this model, and why other models are employed elsewhere. This is generally quite messy. You need a structured introduction upfront in the paper explaining these choices.

**Authors response: see response (1)**

Page 9 - line 31: Again, this is not hazard, probably something else but not hazard... Please revise sentence. *Authors response:* "hazard" replaced by "impact" in the sentence

Page 11 - line 4: Delete double "smaller" Authors response: Done

Page 11 - new paragraph marked red: Not clear what this paragraph add, it is confusing. We have repeatedly shown the effect of dispersion in previous studies. I dont see the need for doing this again, it disrupts the text.

Authors response: It is very challenging to please successive reviewers who does not automatically always share the same point of view. Fourier transform analysis was explicitly

requested by one of the former reviewer, hence this first revised version. We feel logical to keep the successive changes requested throughout the review process to respect this process. Analysis of dispersion has been much shortened compared to the previous version.

Page 11 line 31: This was analysed in more detail first by Løvholt et al. (2008), please notify and provide reference. *Authors response:* Done

Page 13 - line 27: Delete double punctuation. *Authors response:* Done

Page 13 - line 28: Again, this is not hazard assessment, but only an assessment of possible inundation or impact. Please revise title. *Authors response:* done and see response (3)

Page 13 - first three paragraphs of section 3.5: I find all this analytical analysis strange for a phenomena so strongly controlled by local phenomena. Why not limit the impact analysis to the local inundation study. I would suggest to skip this part, and only keep the part using NSW inundation analysis. The paper is overloaded with results, and this is for me a distraction. Moreover, such a rough analytical analysis could be worthwhile for assessing the hazard region, but not for a local analysis.

**Authors response: this aspect has been totally removed from the article.**

Page 15 - line 5: As said above, the authors does not seem to take into account that the dynamics and material behavior is uncertain, and that a simple glass bead experiment cannot be conveyed to real situation. The paragraph should be rewritten to better reflect this. Granted, the simulations fit better the experiments, but the authors have no guarantee that the slope failure will behave this way. Probably it will not.

**Authors response: see response (2)**

Page 15 - line 15: Again, please replace the term hazard assessment with something more appropriate, such as an impact assessment. The study is not broad enough and does not cover return periods, so cannot be coined a hazard study. *Authors response:* replaced by "impact" *and* see response (3)

Page 15 - line 31: This discussion of model effects is too long. I would suggest to shorten it dramatically, as results are shown above and the physics is well-known. Besides, the effects of dispersion have been investigated in previous studies. It can also be analysed with a dispersion number (e.g. Glimsdal et al., 2013)

Authors response: We understand the point of view of the reviewer, but this discussion is justified in the context described at the end of the new introduction (p 3, 1 9) (model comparisons and recommendations). It was also meant to answer the first reviews of the paper. Again, this analysis is now much shorter.

Page 16 - line 23: Wynn and Masson found upward fining, which indicate long separations in time. This means that this was no real retrogression, but more likely separate events. On the other hand, I agree with the authors statement in the last part of this paragraph.

**Authors response:**

The present work did not explicitly take into account the possibility of a retrogressive scenario. Whether the flank collapse occurs en masse or in successive stages is obviously crucial in terms of wave generation. In this study, we proposed several slide volume scenarios which can be used for a crude assessment of the wave reduction in case the collapse occurred as several separate events with no interactions between the successive slides (e.g. the 20 km scenario may give an idea of what would happen if a 80 km slide were occurring progressively or in sequence). The interactions could be left for future research even though field evidences tend to show that these collapses may have occurred as separate events (Wynn and Masson, 2003) rather than in an actual retrogressive way.

Page 17 - line 13: See comment above several times on uncertainty, and reformulate accordingly. *Authors response:* Sentence modified

Initial: The new wave source is reduced in half compared to previous estimations mainly due to the improved rheology calibration

changed to:

The new wave source is reduced in half compared to previous estimations mainly due to the larger value of slide viscosity used in this work

Page 17 - line 20: This sentence is not well formulated, I dont fully understand what you mean. *Authors response:* done

Initial sentence: After 15 minutes of propagation in a Boussinesq model, the wave signal is still dispersive and therefore Boussinesq models should be recommended to use the source provided

modified as: The tsunami source calculated in this paper after 15 minutes of propagation in FUNWAVE-TVD and proposed to the community in the SEANOE repository is dispersive and therefore we recommend to use appropriate models (e.g., Boussinesq models) to propagate further this source in future studies.

Figure 8: Slide contours are very difficult to read. I suggest fewer and larger figures allowing the reader to see the details.

Authors response: The Figure has been split in two figures (Figures 8 and 9) so as to respect the reviewer's wish.

Authors note : in the new manuscript which follows, characters in blue are the remaining changes from the last version, while new changes are marked in red.

**La Palma landslide tsunami: calibrated wave source and assessment of impact in French territories.SA**

Stéphane Abadie1, Alexandre Paris1,2, Riadh Ata3, Sylvestre Le Roy4, Gael Arnaud5, Adrien Poupardin2,6, Lucie Clous1, Philippe Heinrich2, Jeffrey Harris3, Rodrigo Pedreros4, and Yann Krien5 1Universite 
[revised manuscript text omitted]